# Activation of polycystin-1 signaling by binding of stalk-derived peptide agonists

Shristi Pawnikar[1], Brenda S Magenheimer[2,3], Keya Joshi[4], Ericka Nevarez-Munoz[2], Allan Haldane[5], Robin L Maser[2,3,6]*, Yinglong Miao[4]*

[1]Center for Computational Biology and Department of Molecular Biosciences, University of Kansas, Lawrence, United States; [2]Clinical Laboratory Sciences, University of Kansas Medical Center, Kansas City, United States; [3]The Jared Grantham Kidney Institute, University of Kansas Medical Center, Kansas City, United States; [4]Department of Pharmacology and Computational Medicine Program, University of North Carolina, Chapel Hill, United States; [5]Department of Physics, and Center for Biophysics and Computational Biology, Temple University, Philadelphia, United States; [6]Department of Biochemistry and Molecular Biology, University of Kansas Medical Center, Kansas City, United States

*For correspondence:
rmaser@kumc.edu (RLM);
Yinglong_Miao@med.unc.edu
(YM)

Competing interest: The authors declare that no competing interests exist.

**Abstract** Polycystin-1 (PC1) is the protein product of the *PKD1* gene whose mutation causes autosomal dominant Polycystic Kidney Disease (ADPKD). PC1 is an atypical G protein-coupled receptor (GPCR) with an autocatalytic GAIN domain that cleaves PC1 into extracellular N-terminal and membrane-embedded C-terminal (CTF) fragments. Recently, activation of PC1 CTF signaling was shown to be regulated by a stalk tethered agonist (TA), resembling the mechanism observed for adhesion GPCRs. Here, synthetic peptides of the first 9- (p9), 17- (p17), and 21-residues (p21) of the PC1 stalk TA were shown to re-activate signaling by a stalkless CTF mutant in human cell culture assays. Novel Peptide Gaussian accelerated molecular dynamics (Pep-GaMD) simulations elucidated binding conformations of p9, p17, and p21 and revealed multiple specific binding regions to the stalkless CTF. Peptide agonists binding to the TOP domain of PC1 induced close TOP-putative pore loop interactions, a characteristic feature of stalk TA-mediated PC1 CTF activation. Additional sequence coevolution analyses showed the peptide binding regions were consistent with covarying residue pairs identified between the TOP domain and the stalk TA. These insights into the structural dynamic mechanism of PC1 activation by TA peptide agonists provide an in-depth understanding that will facilitate the development of therapeutics targeting PC1 for ADPKD treatment.

## eLife assessment

This joint computational/experimental study demonstrates the ability of synthetic peptides derived from the stalk tethered agonist in polycystin-1 (PC1) to re-activate signaling by a stalkless C-terminal fragment of PC1. The study is **valuable** as it discovered peptide agonists for PC1 and the integrated *in vitro* and *in silico* approach is potentially applicable to the analysis of related systems. Following the revision, the line of evidence presented in the current manuscript is considered **convincing**.

## Introduction

PC1 is the protein product of the *PKD1* gene that is mutated in the majority of cases (~85%) of ADPKD (*Harris and Torres, 2014*). ADPKD is a potentially lethal disease, affecting >0.6 million individuals in the US. It causes renal cyst formation that could consequently lead to kidney failure. Currently, the only approved treatment for ADPKD is Jynarque, a small-molecule antagonist of the arginine vasopressin

receptor 2, V2R, whose signaling, and production of cAMP has been shown to be increased in PKD. This drug targets one of the aberrant pathways downstream from the PKD gene mutation but is inadequate due to its limitations in only slowing disease progression and causing adverse side effects (*Ingelfinger, 2017*). ADPKD severity is dependent on the functional level of PC1, and as such, therapies designed to increase the level of PC1 protein, and its functionality are currently being pursued (*Hopp et al., 2012*; *Cai et al., 2014*; *Hofherr et al., 2016*; *Krappitz et al., 2016*; *Lakhia et al., 2022*). Approximately one-third of *PKD1* mutations are non-truncating and could encode partially functional PC1 protein (*Hopp et al., 2012*; *Tan et al., 2011*; *Rossetti et al., 2009*; *Heyer et al., 2016*). As such, therapeutic treatments that directly target and activate PC1 may represent a promising approach for the treatment of ADPKD. However, this approach remains difficult due to incomplete knowledge of the proximal-most functions of PC1.

PC1 shares characteristics with the Adhesion class of GPCRs (ADGRs), including a conserved GPCR autoproteolysis inducing (GAIN) domain that directs autocatalytic cleavage at an embedded GPCR proteolysis site (GPS) motif (*Araç et al., 2012*). Intramolecular cleavage at the GPS motif generates two non-covalently attached fragments - the extracellular N-terminal fragment (NTF) and the membrane-embedded C-terminal fragment (CTF) (*Qian et al., 2002*; *Kurbegovic et al., 2014*). Similar to ADGRs, the PC1 NTF consists of multiple adhesive domains that promote interactions between cells and with the extracellular matrix (*Kim et al., 2016*; *Weston et al., 2001*; *Sandford et al., 1997*; *Ibraghimov-Beskrovnaya et al., 2000*; *Weston et al., 2003*), while the PC1 CTF is composed of 11 transmembrane (TM) helices and a short C-terminal tail (C-tail) (*Nims et al., 2003*) that has been shown to interact with G proteins for signaling activation or regulation (*Maser and Calvet, 2020*; *Maser et al., 2022*) and has thus led to description of PC1 as an atypical GPCR. Previous studies demonstrated the critical importance of cleavage at the PC1 GPS site to prevent renal cystogenesis in mouse models (*Cai et al., 2014*; *Yu et al., 2007*). For the ADGRs, a TA model has been proposed for activation of G protein signaling. After dissociation of the NTF, the N-terminal stalk of the ADGR CTF interacts with its membrane-embedded TM domains to induce conformational rearrangements that mediate activation of G protein signaling (*Liebscher et al., 2014*; *Schöneberg et al., 2015*; *Demberg et al., 2015*; *Stoveken et al., 2015*). Exogenous synthetic peptides consisting of various lengths of the N-terminal sequence of the stalk have been shown to function as soluble agonists in the activation of signaling by full-length and CTF mutants for numerous ADGRs (*Maser and Calvet, 2020*; *Xiao et al., 2022*).

In previous studies of the PC1 CTF, we revealed a stalk TA-dependent molecular mechanism underlying CTF-mediated activation of an NFAT promoter luciferase reporter through complementary *in vitro* cell signaling experiments and all-atom Gaussian accelerated Molecular Dynamics (GaMD) simulations (*Pawnikar et al., 2022*). GaMD is an unconstrained enhanced sampling method that works by adding a harmonic boost potential to reduce large biomolecular energy barriers (*Miao et al., 2014*) and has been used successfully to capture multiple complex biological processes (*Miao et al., 2015*; *Miao and McCammon, 2016*; *Pang et al., 2017*; *Miao and McCammon, 2017*; *Wang and Chan, 2017*; *Liao and Wang, 2019*; *Miao et al., 2018*; *Chuang et al., 2018*; *Sibener et al., 2018*; *Park et al., 2018*; *Miao and McCammon, 2018*; *Ricci et al., 2019*; *Palermo et al., 2017*; *Bhattarai et al., 2020*; *Pawnikar and Miao, 2020*) including GPCR activation (*Miao and McCammon, 2016*). Expression constructs encoding a stalkless PC1 CTF (a nonbiological mutant with deletion of the first 21 N-terminal residues of CTF) and three ADPKD-associated missense mutants within the stalk region (G3052R, R3063C, and R3063P) were shown to be defective in reporter activation as compared to wild-type PC1 CTF. GaMD simulations revealed a novel allosteric transduction pathway for activation of PC1 CTF signaling that involves initiation by the Stalk interacting with a large extracellular loop between TM segments S1/TM6 and S2/TM7, called the TOP domain, followed by close interactions between the TOP and a putative pore loop (PL) domain between the final 2 TM domains. GaMD simulations of the wild-type PC1 CTF also identified a 'Closed/Active' low-energy state related to the large number of Stalk-TOP contacts and the R3848-E4078 ionic interaction between the TOP and PL domains that was not present in the stalkless CTF (*Pawnikar et al., 2022*).

Here, we have utilized *in vitro* cell signaling assays to identify peptide agonists targeting PC1 in combination with *in silico* studies to investigate their binding mechanisms for activation of PC1 signaling. Synthetic peptides of 7–21 residues in length derived from the N-terminus of the PC1 CTF stalk sequence were tested for their ability to re-activate signaling of the stalkless CTF expression construct. Peptide docking and simulations with the recently developed Peptide GaMD (Pep-GaMD),

which is able to characterize peptide-protein binding processes more efficiently (*Wang and Miao, 2020*), were combined for selected peptide agonists p9, p17, and p21 to gain insight into their binding mechanism to the stalkless PC1 CTF. Pep-GaMD was able to successfully refine the docking conformations of the peptides bound to the extracellular TOP domain of PC1. In further Pep-GaMD simulations, the key salt bridge interaction between R3848 and E4078 from the TOP domain and PL, respectively, was observed upon binding of the peptides to stalkless PC1 CTF. Using Potts covariation analysis, in which a protein fitness model is inferred based on observed mutational covariation patterns in multiple sequence alignments (MSAs) of homologous proteins (*Levy et al., 2017*), we identified residues in the PC1 stalk with direct mutational covariation with residues in the TOP domain, which were strikingly consistent with the binding interfaces identified in docking and simulation studies. Overall, these analyses yielded mechanistic insights underlying the stalk peptide agonist-mediated signal re-activation of stalkless PC1 CTF. Such insights provide significant contributions toward the future design and development of peptide modulators targeting PC1 for an effective ADPKD therapeutic treatment.

## Results

### Synthetic, stalk-derived peptides re-activate NFAT reporter by CTF$^{\Delta st}$ *in trans*

Our previous study utilized expression constructs of human PC1 CTF. However, in order to prepare for eventual *in vivo* experiments in mouse models, we generated expression constructs of mouse (m) PC1 consisting of the signal peptide sequence of the T cell surface glycoprotein CD5 (MPMG SLQPLATLYLLGMLVASVLG) fused in frame with the stalk sequence of wild-type mCTF beginning with residue T3041, or with a 'stalkless' CTF lacking the first 21 residues of the stalk (mCTF$^{\Delta st}$) beginning with residue S3062 (*Figure 1A*). The CD5 signal peptide coding sequence was added to the wild-type mCTF and stalkless mCTF$^{\Delta st}$ expression constructs in order to ensure their translation at the endoplasmic reticulum for plasma membrane localization.

Transient transfection of HEK293T cells with either empty expression vector (ev), CTF or CTF$^{\Delta st}$ showed that the CTF$^{\Delta st}$ mutant exhibited a dramatic loss of NFAT reporter activation that was essentially reduced to ev control levels (*Figure 1B*). Both total (*Figure 1C–D*) and cell surface (*Figure 1—figure supplement 1A–B*) expression levels of CTF$^{\Delta st}$ were comparable to CTF, which suggests that neither protein stability nor membrane trafficking was responsible for the inability of CTF$^{\Delta st}$ to activate the NFAT reporter. These results are consistent with those obtained using expression constructs of human PC1 that demonstrated the stalk region of PC1 CTF acts as a tethered peptide agonist (*Pawnikar et al., 2022*).

To further investigate the agonistic property of the CTF stalk, we synthesized peptides (p) consisting of the N-terminal 7, 9, 11, 13, 15, 17, 19, or 21 residues from the stalk sequence of mPC1. All peptides were appended with a C-terminal, 7-residue hydrophilic sequence (GGKKKKK) to increase solubility. HEK293T cells were transiently transfected with empty expression vector or mCTF$^{\Delta st}$ plasmids along with the NFAT luciferase reporter and then treated with stalk peptides p7 through p21 or with the addition of culture medium only ('no peptide' control). The NFAT reporter was significantly activated in CTF$^{\Delta st}$-transfected cells by treatment with p7, p9 or p17 as compared to their corresponding ev + peptide treatment controls. These stalk peptides also significantly increased reporter activity in comparison to the CTF$^{\Delta st}$ with no peptide treatment control (*Figure 1E*). Treatment of CTF$^{\Delta st}$-transfected cells with p19 or p21 also significantly increased reporter activation in comparison to the CTF$^{\Delta st}$ + no peptide control; however, reporter activation occurred in both ev- and CTF$^{\Delta st}$-transfected cells treated with either p19 or p21, suggesting that p19- and p21-mediated activation was not dependent on exogenous expression of mouse CTF$^{\Delta st}$ and could be activating the endogenous human PC1 protein. Treatment of ev- and CTF$^{\Delta st}$-transfected cells with a peptide consisting of the hydrophilic sequence alone (i.e., solubility peptide) showed that the solubility tag was not responsible for the rescue of CTF$^{\Delta st}$-mediated reporter activation by stalk peptides such as p17 (*Figure 1—figure supplement 2*). Altogether, these results were consistent with soluble stalk-derived peptides acting as PC1 CTF agonists *in trans*, and provided additional support for the PC1 CTF stalk region harboring TA activity (*Pawnikar et al., 2022*). We hypothesized that the soluble, activating peptides bind to the TOP domain of PC1 in a manner mimicking the tethered stalk in order to reactivate the

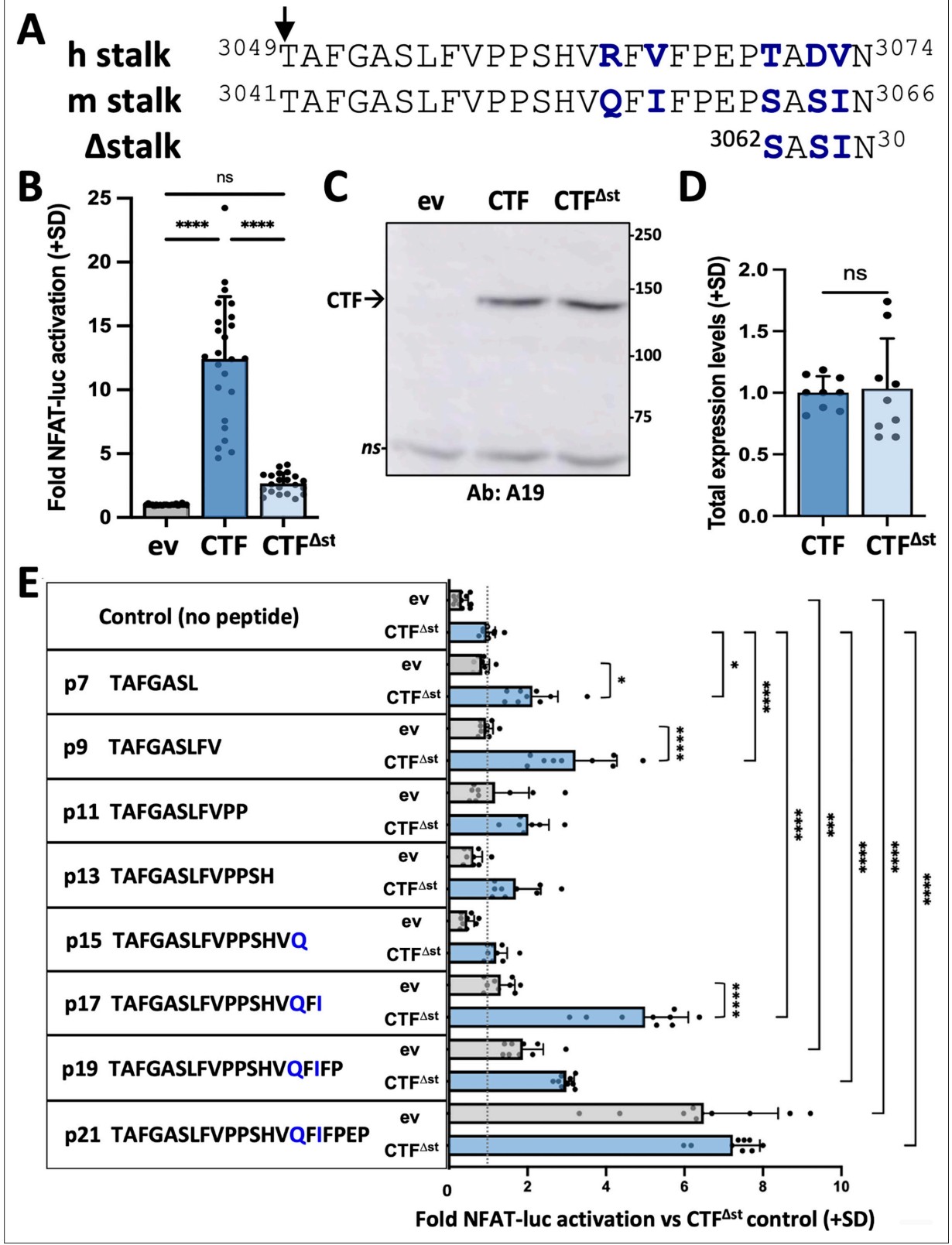

**Figure 1.** Synthetic peptides derived from the stalk sequence of polycystin-1 (PC1) can stimulate the signaling of stalkless PC1 C-terminal fragment (CTF). (**A**) Alignment of CTF stalk sequences from human (**h**) and mouse (**m**) PC1. CTF$^{\Delta st}$ has a 21-residue deletion from the N-terminal end of the stalk region. Arrow, GPCR proteolysis site (GPS) cleavage site. Non-identical residues are shown in bolded blue. (**B**) Activation of the NFAT-luc reporter by transfected mCTF or mCTF$^{\Delta st}$ expression constructs shown relative to empty expression vector (ev) as means (+ standard deviation, SD) of three wells/

*Figure 1 continued on next page*

*Figure 1 continued*

construct from each of seven independent experiments. (**C**) Representative Western blot of total cell lysates from one of the experiments in (**B**), probed with antisera A19 against mouse PC1 C-tail. ns, non-specific. (**D**) Summary of the total expression levels (means + SD) of CTF$^{\Delta st}$ relative to CTF from the experiments in (**B**). (**E**) Stalk peptide treatment of expression vector (ev)- or mCTF$^{\Delta st}$-transfected cells. Sequences of stalk-derived peptides p7-p21 are shown. Graph represents the fold NFAT-luc activation for both ev- (gray bars) and CTF$^{\Delta st}$- (blue bars) transfected cells relative to the CTF$^{\Delta st}$ control after 24 hr treatment with or without peptide. Results are the means (+ SD) of three separate experiments, each with three wells/conditions. *p<0.05; ***p=0.0001; ****p<0.0001. Analysis by one-way ANOVA with Tukey-Kramer post-test.

The online version of this article includes the following source data and figure supplement(s) for figure 1:

**Source data 1.** Uncropped and labeled Western blot data for *Figure 1C*.

**Source data 2.** Raw unedited Western blot data for *Figure 1C*.

**Figure supplement 1.** Cell surface expression of mouse polycystin-1 (PC1) C-terminal fragment (CTF), and mCTF$^{\Delta stk}$.

**Figure supplement 1—source data 1.** Uncropped and labeled Western blot data for *Figure 1—figure supplements 1 and 2*.

**Figure supplement 1—source data 2.** Raw unedited Western blot data for *Figure 1—figure supplements 1 and 2*.

**Figure supplement 2.** Solubility tag peptide treatment of ev- or CTF$^{\Delta st}$-transfected cells.

signaling of the stalkless CTF$^{\Delta st}$ mutant. From among the active stalk-derived peptides, we selected p9, p17, and p21 that exhibited the highest agonist activity in CTF$^{\Delta st}$ reporter activation (*Figure 1E*) for *in silico* simulation studies.

## Docking and Pep-GaMD simulations of peptide agonist binding to stalkless PC1 CTF

We chose to use the stalkless CTF (ΔStalk CTF) as representing the least complex system in which the binding of exogenous peptides could be studied. ΔStalk CTF is not a biological form or a mutant protein of PC1 observed in ADPKD. However, in our previous study, it mimicked the ADPKD-associated stalk mutants by being defective in cell signaling assays and rarely formed the R3848-E4078 salt bridge that was frequently seen in GaMD simulations with wild-type CTF (*Pawnikar et al., 2022*). Therefore ΔStalk CTF served as the negative control for the following studies.

The cryo-EM structure of the human PC1-PC2 complex (PDB: 6A70) (*Su et al., 2018*) was used to build the computational model for ΔStalk PC1 CTF after deleting the first 21 residues (3049–3069) from the CTF (*Pawnikar et al., 2022*). We successfully docked the p9, p17, and p21 stalk peptides to the ΔStalk CTF model with HPEPDOCK (*Zhou et al., 2018*) (See Materials and Methods). The peptides are all bound to the TOP domain and the interface between the TOP domain and extracellular loop 1 (ECL1) of CTF (*Figure 2—figure supplement 1A–B*). In particular, peptide p21 occupied a closely similar binding region as the stalk in wild-type CTF as observed in the previous study (*Pawnikar et al., 2022*). We then performed five independent 500 ns Pep-GaMD simulations on each of the three stalk peptide agonists p9, p17 and p21 bound to ΔStalk CTF to refine their HPEPDOCK docking conformations (See Materials and Methods).

With the Pep-GaMD simulation frames, we performed structural clustering of each peptide using the hierarchical agglomerative algorithm in CPPTRAJ (*Roe and Cheatham, 2013*). The top-ranked conformations of each peptide bound to ΔStalk CTF were compared to their initial docking conformations. Next, we calculated 2D free energy profiles of the peptides-bound ΔStalk CTF by reweighting the Pep-GaMD simulations. The R3848-E4078 residue distance and the number of contacts between the peptides (p9, p17, and p21) and the TOP domain were selected as the reaction coordinates. The number of contacts was calculated between any atom pairs within 4 Å distance of the peptide and extracellular domains of the PC1 protein. In the subsequent analyses, stalk and peptide residues are numbered relative to the N terminus of the stalk as starting from 1, while residues of the Δstalk CTF are numbered according to the human PC1 protein sequence.

## Active conformation of peptide p9-bound PC1 CTF

From the free energy profile of the p9-bound ΔStalk CTF, we identified 'Unbound' and 'Bound' low-energy states (*Figure 2A*). In the docking conformation, peptide p9 bound to the interface between the TOP and ECL1 of ΔStalk CTF (*Figure 2—figure supplement 1A–B*). In Pep-GaMD simulations, the p9 peptide dissociated from the TOP-ECL1 binding pocket and rebound to the TOP domain in a slightly different region (*Figure 2B–C*). The p9 peptide sequence is mostly composed of hydrophobic

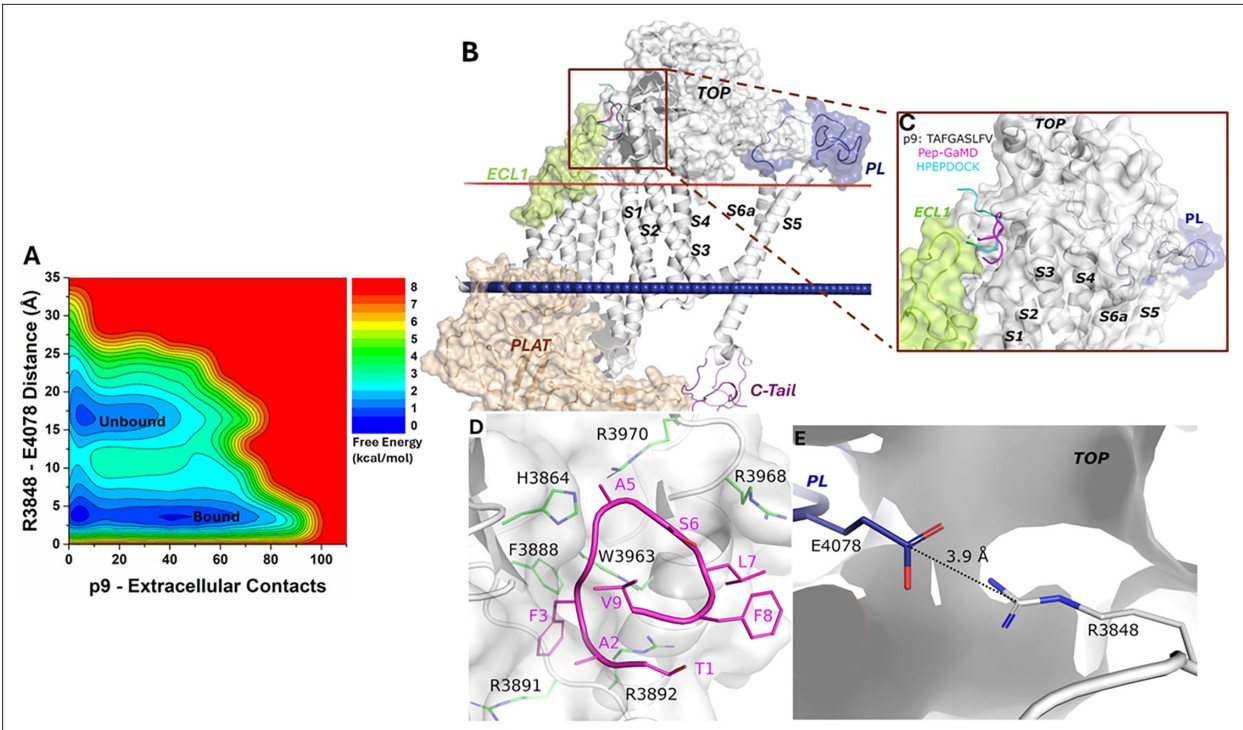

**Figure 2.** Free energy profile and low-energy conformations of the p9-bound ΔStalk CTF obtained from Pep-GaMD simulations. (**A**) Free energy profile of the p9-bound ΔStalk CTF regarding the number of atom contacts between p9 and extracellular domains of CTF and the distance between the CZ atom of R3848 and the CD atom of R4078 in CTF. (**B–C**) Comparison of HPEPDOCK docking (cyan) and Pep-GaMD refined (magenta) conformations of peptide p9 in ΔStalk CTF. (**D**) Polar interactions between peptide-protein residues were observed in the top-ranked representative conformations of p9. Peptide residues are numbered relative to the N terminus of the stalk with the peptide starting from 1, while residues within ΔStalk CTF are numbered according to the standard polycystin-1 (PC1) residue number. (**E**) Distance between the TOP domain residue R3848 and pore loop (PL) residue E4078 observed in p9-bound ΔStalk CTF.

The online version of this article includes the following figure supplement(s) for figure 2:

**Figure supplement 1.** Peptide docking conformations and computational model for Peptide GaMD (Pep-GaMD) simulations.

**Figure supplement 2.** Contact maps showing residue-pairs in contact (black squares) between the peptide (y-axis) and the extracellular domains of C-terminal fragment (CTF) (x-axis), in the representative 'Bound' state for p21 (top), p17 (middle), and p9 (bottom).

**Figure supplement 3.** 2D free energy profiles of the p9 system regarding the number of atom contacts between the p9 and protein extracellular domains and the R3848-E4078 distance (the CZ atom in R3848 and the CD atom in E4078) calculated from (**A**) Sim1, (**B**) Sim2, (**C**) Sim3, (**D**) Sim4, and (**E**) Sim5 of the Peptide GaMD (Pep-GaMD) simulations.

residues. Polar interactions between the main chain atoms of peptide-protein residues were observed in the top-ranked representative conformation of the p9-bound ΔStalk CTF. Protein residues R3892 and H3864 formed hydrogen bonds with p9 residues A2 and A5, respectively (*Figure 2D*). These interactions were also highlighted in the protein contact map between peptide p9 and the extracellular domains of CTF in the representative 'Bound' state (*Figure 2—figure supplement 2*). The distance between the TOP domain residue R3848 and PL residue E4078 was 3.9 Å (*Figure 2E*), suggesting that the top-ranked representative conformation of the p9-bound ΔStalk CTF was in the 'Closed/Active' low-energy state. In addition, the 2D free energy profile of each individual simulation was calculated and the 'Bound' low-energy state was consistently identified in the 2D free energy profiles of peptide p9 (*Figure 2—figure supplement 3*).

## Active and intermediate conformational states of peptide p17-bound PC1 CTF

The free energy profile of the p17-bound ΔStalk system allowed us to identify three low-energy states - 'Unbound,' 'Intermediate,' and 'Bound' (*Figure 3A*). In the docking conformation, peptide p17 bound to the interface between the TOP and ECL1 of ΔStalk CTF (*Figure 2—figure supplement 1A–B*). In

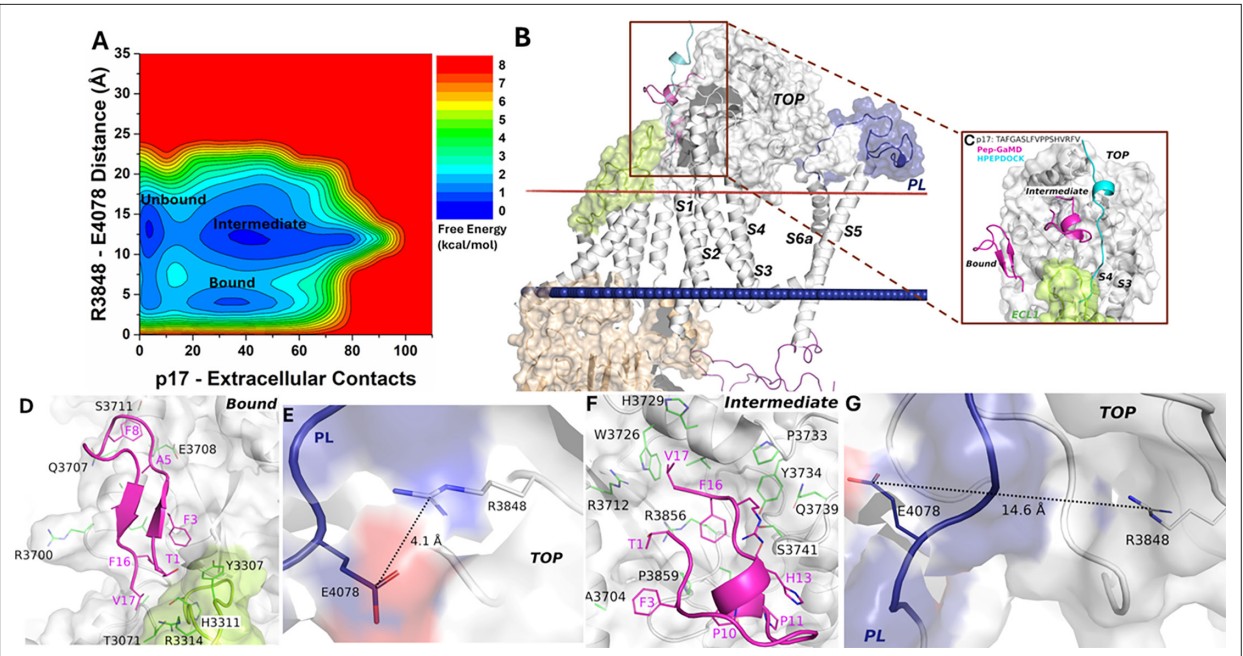

**Figure 3.** Free energy profile and low-energy conformations of the p17-bound ΔStalk CTF obtained from Pep-GaMD simulations. (**A**) Free energy profile of the p17-bound ΔStalk CTF regarding the number of atom contacts between p17 and extracellular domains of CTF and the distance between the CZ atom of R3848 and the CD atom of R4078 in CTF. (**B–C**) Comparison of HPEPDOCK docking (cyan) and Pep-GaMD refined (magenta) conformations of peptide p17 in ΔStalk CTF. Hydrophobic interactions (red dashed lines) between peptide-protein residues were observed in the (**D**) 'Bound' and (**F**) 'Intermediate' low-energy conformations of p17-bound ΔStalk CTF. Distance between the TOP domain residue R3848 and pore loop (PL) residue E4078 observed in the (**E**) 'Bound' and (**G**) 'Intermediate' low-energy conformations of p17-bound ΔStalk CTF.

The online version of this article includes the following figure supplement(s) for figure 3:

**Figure supplement 1.** 2D free energy profiles of the p17 system regarding the number of atom contacts between the p17 and protein extracellular domains and the R3848-E4078 distance (the CZ atom in R3848 and the CD atom in E4078) calculated from (**A**) Sim1, (**B**) Sim2, (**C**) Sim3, (**D**) Sim4, and (**E**) Sim5 of the Peptide GaMD (Pep-GaMD) simulations.

the Pep-GaMD refined 'Bound' state, a folded antiparallel ß-strand conformation was observed for the peptide p17 at the interface of ECL1 and the TOP domain (*Figure 3B–C*). Peptide residues T1, F3, A5, F8, F16, and V17 formed hydrophobic interactions with the protein residues H3311, R3314 and Y3307 from ECL1, and E3708, S3711, Q3707, A3704, R3700, and L3701 from the TOP domain (*Figure 3D*). These interactions were also highlighted in the protein contact map between peptide p17 and the extracellular domains of CTF in the representative 'Bound' state (*Figure 2—figure supplement 2*). The distance between the TOP domain residue R3848 and PL residue E4078 was 4.1 Å (*Figure 3E*), suggesting that the top-ranked representative conformation of the p17 bound ΔStalk CTF was in the 'Closed/Active' low-energy state.

In the 'Intermediate' state, p17 with a short helical turn was also observed to bind the TOP domain of ΔStalk CTF (*Figure 3B–C*). Hydrophobic residue interactions were also formed between the peptide and protein. In particular, peptide residues T1, F3, P10, P11, H13, R15, F16, and V17 formed hydrophobic interactions with the protein residues P3859, A3704, S3741, Q3739, Y3734, P3733, H3729, W3726, R3712, and R3856 from the TOP domain (*Figure 3F*). The distance between the TOP domain residue R3848 and PL residue E4078 was 14.6 Å (*Figure 3G*), suggesting that this representative conformation (ranked the second among the Pep-GaMD structural clusters) of the p17-bound ΔStalk CTF was in the 'Intermediate' low-energy state. In addition, the 2D free energy profile of each individual simulation was calculated. Pep-GaMD simulations were able to refine the peptide conformation from the 'Unbound' to 'Intermediate,' and 'Bound' states in Sim1 and Sim5, while the peptide reached only the 'Intermediate' state in the other three simulations (*Figure 3—figure supplement 1*). The free energy values of 2D PMF minima shown in *Figure 3A* could differ from those in the 1D PMF minima of peptide structural clusters, especially with the usage of distinct reaction coordinates.

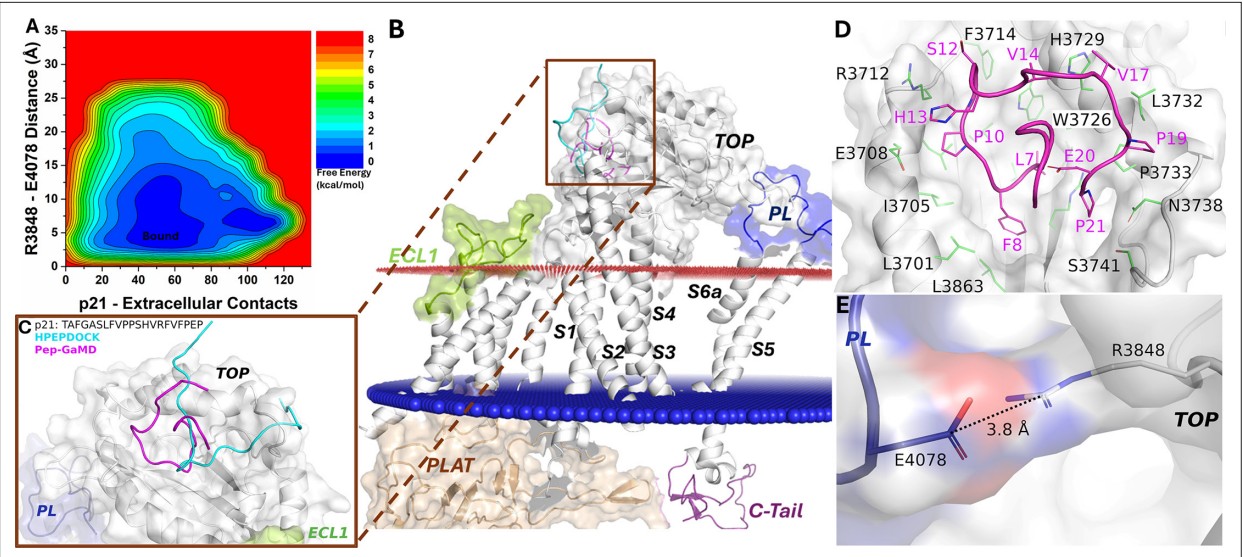

**Figure 4.** Free energy profile and low-energy conformations of the p21-bound ΔStalk CTF obtained from Pep-GaMD simulations. (**A**) Free energy profile of the p21-bound ΔStalk CTF regarding the number of atom contacts between p21 and extracellular domains of CTF and the distance between the CZ atom of R3848 and the CD atom of R4078 in CTF. (**B–C**) Comparison of HPEPDOCK docking (cyan) and Pep-GaMD refined (magenta) conformations of peptide p21 in ΔStalk CTF. (**D**) Polar interactions between peptide-protein residues were observed in the top-ranked representative conformations of p21. (**E**) Distance between the TOP domain residue R3848 and pore loop (PL) residue E4078 observed in p21-bound ΔStalk CTF.

The online version of this article includes the following figure supplement(s) for figure 4:

**Figure supplement 1.** Time courses of the p21 system obtained from the five Peptide GaMD (Pep-GaMD) simulations.

**Figure supplement 2.** 2D free energy profiles of the p21 system regarding the number of atom contacts between the p21 and protein extracellular domains and the R3848-E4078 distance (the CZ atom in R3848 and the CD atom in E4078) calculated from (**A**) Sim1, (**B**) Sim2, (**C**) Sim3, (**D**) Sim4, and (**E**) Sim5 of the Peptide GaMD (Pep-GaMD) simulations.

## Active conformational state of peptide p21-bound PC1 CTF

Finally, the free energy profile of the p21-bound ΔStalk CTF allowed us to identify only a broad low-energy well corresponding to the 'Bound' state (*Figure 4A*). The docking conformation of p21-bound ΔStalk CTF was refined through Pep-GaMD simulations, where folding of the peptide was observed on the protein surface of the TOP domain (*Figure 4B–C*). The p21 peptide occupied a similar binding region as the stalk in wild-type CTF as observed in the previous study (*Pawnikar et al., 2022*). Hydrophobic contacts were observed between peptide residues L7, F8, P10, S12, H13, V14, V17, P19, E20, and P21 and protein residues L3863, L3701, I3705, L3709, E3708, R3712, F3714, H3729, W3726, V3730, L3732, P3733, N3738, R3856, and S3741 (*Figure 4D*). These interactions were also highlighted in the protein contact map between peptide p21 and the extracellular domains of CTF in the representative 'Bound' state (*Figure 2—figure supplement 2*). The distance calculated from the top-ranked structural cluster of the system between the TOP domain residue R3848 and PL residue E4078 was 3.8 Å, corresponding to the 'Closed/Active' low-energy state (*Figure 4E*). Furthermore, time courses of the radius of gyration (*Rg*) and root-mean-square deviation (RMSD) of p21 relative to the starting HPEPDOCK conformation showed large conformational changes of the peptide during Pep-GaMD simulations (*Figure 4—figure supplement 1A–C*). In addition, the 2D free energy profile of each individual simulation showed that Pep-GaMD was able to refine the peptide docking conformation to the 'Bound' state in all the five simulations (*Figure 4—figure supplement 2*).

## Peptide binding regions correlated with covarying residue pairs identified between the TOP domain and stalk TA

To provide an independent basis of evidence supporting the observation of 'Bound' and 'Intermediate' states of agonist peptide binding, we constructed a multiple sequence alignment (MSA) with an effective count of 1022 evolutionarily diverged PC1 homologs (illustrated in *Figure 5—figure supplement 1*) from which we inferred a Potts statistical model (*Figure 5*). Columns of the MSA with

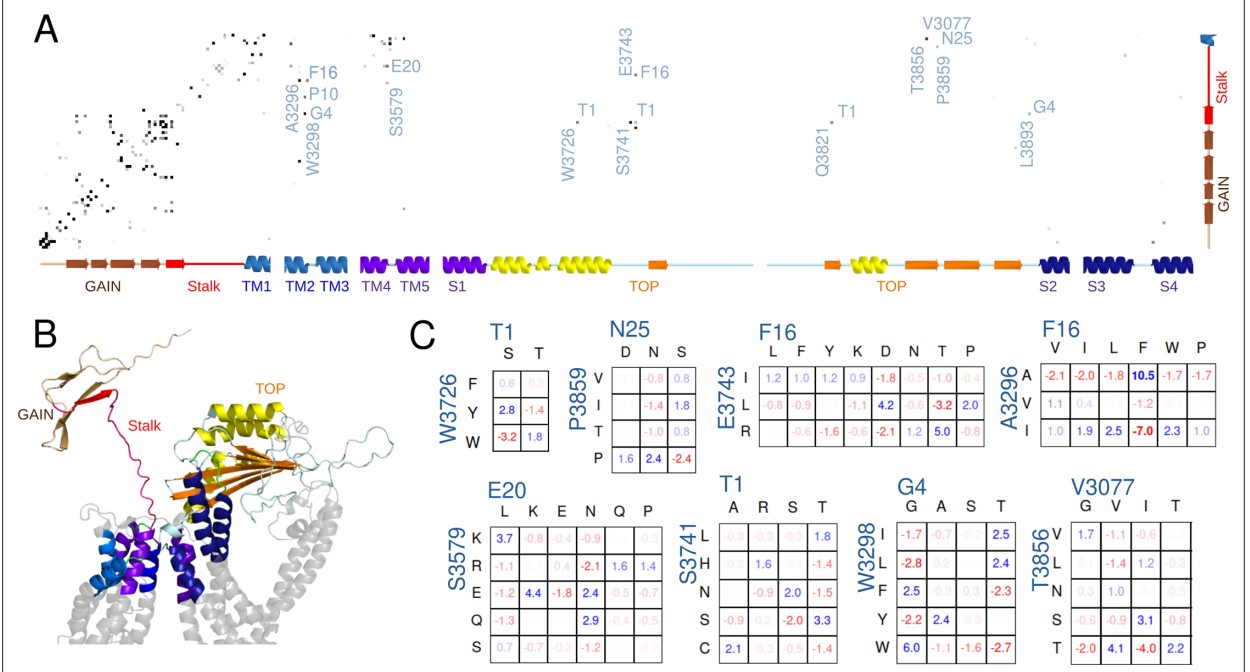

**Figure 5.** Potts covariation analysis of residue interactions in polycystin-1 (PC1). (**A**) Potts interaction map based on the Polycystic Kidney Disease 1 (*PKD1*) multiple-sequence-alignment illustrated in *Figure 5—figure supplement 2*, showing interactions with the stalk. Gray dots are shown for residue position-pairs with Potts covariation scores above a threshold, colored darker for higher scores, and selected interacting pairs are annotated with the stalk residue (horizontal, numbered from the stalk N-terminus) and other residue (vertical, standard PC1 numbering) with the PC1 residue at each position. The secondary structure as a function of position is annotated along the axes. (**B**) Cartoon showing the subset of PC1 included in the Potts covariation analysis colored as in the secondary structure in panel A, using a structure predicted by AlphaFold. Gray regions were excluded from the Potts model. (**C**) Residue Covariation scores for selected position-pairs. The scores reflect the percentage excess frequency of the residue-pair relative to the null expected frequency if the multiple sequence alignment (MSA) columns were uncorrelated, with blue values reflecting excess and red dearth. Only the most common residue types are shown.

The online version of this article includes the following figure supplement(s) for figure 5:

**Figure supplement 1.** Sequence alignment of Polycystic Kidney Disease 1 (*PKD1*) homologs showing the 394-residue extracellular region included in the Potts model.

**Figure supplement 2.** Comparison of residue-residue interactions within the end of the polycystin-1 (PC1) GPCR autoproteolysis inducing (GAIN) domain.

'direct' statistical interactions, as detected using the Potts inference method, reflect compensatory mutation pairs maintained through evolution supporting a conserved function. We limited our MSA to 394 residues on the extracellular side of PC1 because of the computational challenge of fitting the entire PC1 sequence (*Figure 5B* and *Figure 5—figure supplement 1*).

*Figure 5A* shows the pairs of positions with strong Potts interaction scores where one position is either in the stalk, the nearby GAIN domain, or the TM1 helix. Some predicted interaction pairs recapitulated beta-sheet contacts within the GAIN domain observed in the homolog rat latrophilin-1 (*Araç et al., 2012*) as well as predicted by Alphafold (*Jumper et al., 2021*, *Figure 5—figure supplement 2*) or between the extracellular ends of the TM2/3 and TM4/5 alpha helices known from cryo-EM structures (*Su et al., 2018*) or predicted by Alphafold, validating that our model detected biologically functional interactions.

We identified strong interactions between the stalk and other residues from the Potts model. They were not observed in the cryo-EM structure, in which the flexible stalk is missing. For interactions with the TOP domain, out of the 4875 possible pairs (25 stalk residues by 195 TOP domain residues in our Potts model), this analysis detected a stringent set of 6 strongly interacting pairs. Remarkably, multiple positions in this small set were among those relevant to the 'Intermediate' binding conformation of p17 and 'Bound' conformation of p21 as identified from the Pep-GaMD simulations. These were W3726 and S3741 in the TOP domain, both interacting with T1 of the stalk, and P3859 interacting

with N25 at the end of the stalk (*Figure 5A*). Additionally, we identified E3743 to be strongly interacting with F16 in the stalk, and it was also near the observed binding region in the TOP domain for the peptide p17 in the 'Intermediate' state near S3741 (*Figure 3F*). The remaining two strong interactions between the stalk and the TOP domain involved Q3821 with T1 and L3893 with G4. L3893 is adjacent to R3892 that was identified to interact with the peptide p9 in the 'Bound' state (*Figure 2D*) and mutating it may affect its neighbor's positioning. Besides the interactions between the TOP domain and the stalk TA, we also found a set of interactions between the stalk and the extracellular ends of TM2-TM3 helices and TM4-TM5 helices, in which stalk residues G4, P10, F16, and E20 interact with W3298, A3296, and S3579, respectively, as well as a strong interaction between V3077, three residues past the end of the stalk, and T3856 in the TOP domain. TOP domain residue T3856 was also identified as relevant to the binding region of peptide p17 in the 'Intermediate' state (*Figure 3F*) and peptide p21 in the 'Bound' state (*Figure 4D*). These interactions could additionally play a role in stalk-TA activation or could be related to other functionality such as cleavage in the GPS motif.

To gain further insight and to validate that these detected 'direct' interactions reflect biologically meaningful functional interactions and are not artifacts of the data, we examined the residue-specific covariation observed in the MSA (*Figure 5C*), which measures the difference between the observed pairwise residue frequency and its null expectation under the assumption of independent variation. Values greater than ~1% are commonly found to be indications of a statistically reliable mutational covariation (see Materials and Methods), and many of the covarying pairs discovered involving the stalk were significantly above this value. We validated that the covarying residue pairings were consistent with biophysical interaction. For example, for the position-pair 20–3579, here annotated such that the first index is the stalk residue numbered relative to the N terminus of the stalk and the second index is the TOP domain residue numbered according to the human PC1 protein sequence, there were excess residue-pair counts in the MSA consistent with opposite-charge or polar pairing such as K20-E3579, N20-Q3579, and others, and a dearth of repulsive like-charge pairs such as E20-E3579. Similarly, position-pair 1–3741 favored certain combinations of polar residues such as T1-S3741. Other position-pairs appeared consistent with hydrophobic packing interactions, such as F16-A3296, G4-W3298, and T1-W3726. A large residue F or W at position 3298 in the extracellular end of TM2 was commonly paired with a G at stalk position 4, while a smaller I or L residue at position 3298 was more commonly paired with a T at stalk position 4.

Molecular Mechanics/Poisson-Boltzmann Surface Area (MM/PBSA) (*Rastelli et al., 2010*) analysis was further performed to calculate the binding free energies of peptides p9, p17, and p21 to PC1 CTF and decompose the residue-wise energy contributions using the gmx_MMPBSA software (*Valdés-Tresanco et al., 2021*). The relative rank of the overall peptide binding free energies (*Table 1*) was consistent with the

**Table 2.** Summary of residue-wise energy decomposition analysis between the peptide p9 and polycystin-1 (PC1) C-terminal fragment (CTF) in the bound state sampled during Peptide GaMD (Pep-GaMD) simulations.

| Residue | ΔG (kcal/mol) |
|---|---|
| T1 | −10.47±6.92 |
| A2 | −3.28±2.18 |
| F3 | −2.62±2.62 |
| G4 | −0.30±2.38 |
| A5 | −1.61±2.67 |
| S6 | −0.59±4.49 |
| L7 | −0.28±2.96 |
| F8 | −1.51±1.84 |
| V9 | −10.14±5.68 |
| R3891 | −0.14±7.26 |
| R3892 | −0.31±7.85 |
| F3888 | −3.24±2.95 |
| H3864 | −0.03±3.86 |
| R3970 | −0.94±2.62 |
| R3968 | −0.27±1.67 |

**Table 1.** Summary of MM/PBSA binding free energy analysis for the peptides p9, p17 and p21 and polycystin-1 (PC1) C-terminal fragment (CTF) in the bound state sampled during Peptide GaMD (Pep-GaMD) simulations.

| System | ΔG (kcal/mol) |
|---|---|
| p21 | −40.29±6.94 |
| p9 | −17.30±4.50 |
| p17 | −12.74±5.62 |

**Table 3.** Summary of residue-wise energy decomposition analysis between the peptide p17 and polycystin-1 (PC1) C-terminal fragment (CTF) in the bound state sampled during Peptide GaMD (Pep-GaMD) simulations.

| Residue | ΔG (kcal/mol) |
|---|---|
| T1 | −10.98±2.32 |
| A2 | −0.36±2.34 |
| F3 | −0.23±3.82 |
| G4 | −0.03±3.83 |
| A5 | −0.86±3.08 |
| S6 | −0.48±4.06 |
| L7 | −0.05±2.90 |
| F8 | −0.21±4.42 |
| V9 | −0.19±3.41 |
| P10 | −0.72±2.64 |
| P11 | −0.09±3.23 |
| S12 | −0.29±4.92 |
| H13 | −1.19±7.37 |
| V14 | −1.25±3.08 |
| R15 | −10.63±3.76 |
| F16 | −0.93±5.52 |
| V17 | −10.20±3.83 |
| Y3307 | −0.11±2.68 |
| H3311 | −0.04±3.25 |
| R3314 | −8.90±1.65 |
| R3700 | −14.63±7.8 |
| Q3707 | −0.27±4.96 |
| S3711 | −0.08±4.06 |
| E3708 | −10.14±4.15 |

**Table 4.** Summary of residue-wise energy decomposition analysis between the peptide p21 and polycystin-1 (PC1) C-terminal fragment (CTF) in the bound state sampled during Peptide GaMD (Pep-GaMD) simulations.

| Residue | ΔG (kcal/mol) |
|---|---|
| T1 | −0.25±1.66 |
| A2 | −1.02±2.90 |
| F3 | −0.08±0.10 |
| G4 | −2.42±2.65 |
| A5 | −0.11±2.15 |
| S6 | −0.09±1.31 |
| L7 | −0.83±1.43 |
| F8 | −0.68±1.10 |
| V9 | −8.17±2.10 |
| P10 | −6.16±1.79 |
| P11 | −1.22±2.83 |
| S12 | −0.05±2.56 |
| H13 | −1.13±3.74 |
| V14 | −1.16±1.78 |
| R15 | −0.59±1.02 |
| F16 | −0.59±1.02 |
| V17 | −1.59±1.13 |
| F18 | −0.20±1.37 |
| P19 | −7.09±1.51 |
| E20 | −3.76±2.24 |
| P21 | −8.43±2.01 |
| L3863 | −0.33±2.28 |
| L3701 | −0.04±2.39 |
| I3705 | −0.51±2.16 |
| E3708 | −1.68±3.11 |
| R3712 | −3.85±1.56 |
| F3714 | −0.01±2.28 |
| W3726 | −2.93±2.11 |
| H3729 | −0.56±2.00 |
| L3732 | −0.26±2.49 |
| P3733 | −0.61±2.33 |
| N3738 | −0.07±4.91 |
| S3741 | −0.03±4.27 |

experimental signaling data, i.e., p21 > p9 > p17, for which p21 showed the largest free energy value of binding (−40.29±6.94 kcal/mol). Furthermore, we performed residue-wise energy decomposition analysis with MM/PBSA using gmx_MMPBSA software (*Valdés-Tresanco et al., 2021*), which allowed us to identify key residues that contributed the most to the peptide binding energies. These included residues T1 and V9 in p9 (*Table 2*), residues T1, R15 and V17 in p17 (*Table 3*), and residues P10, P11, P19, and P21 in p21 and residue W3726 in the PC1 CTF (*Table 4*). The energetic contributions of these residues apparently correlated to the sequence coevolution predicted from the Potts model.

# Discussion

In *in vitro*, cell-based signaling assays, PC1 CTF-mediated activation of the NFAT reporter is dependent on its N-terminal, extracellular stalk, as shown by the loss of reporter activity with the CTF stalk-deletion expression construct, CTF$^{\Delta st}$ (*Pawnikar et al., 2022*, *Figure 1B*) and by the ability of synthetic, stalk sequence-derived peptides to reactivate signaling by CTF$^{\Delta st}$ *in trans* (*Figure 1E*). A series of synthetic peptides derived from the N-terminal sequence of the mouse PC1 CTF stalk were used to determine their agonistic activity in PC1 CTF$^{\Delta st}$-transfected cells. Notably, treatment with stalk peptides p7, p9, p17, p19, and p21 resulted in significant NFAT reporter activity over CTF$^{\Delta st}$ control (no peptide treatment), wherein the effects of p7, p9, and p17 were specific to mouse CTF$^{\Delta st}$-expressing cells. These data are consistent with the stalk peptides acting as soluble TA peptide agonists for PC1 and provide further evidence for the activation of PC1 signaling via an ADGR-like TA mechanism.

In ADPKD, numerous missense mutations reported within the GAIN domain that have been shown to prevent or inhibit cleavage at the GPS (*Araç et al., 2012*). Loss of PC1 GPS cleavage, which is known to cause ADPKD, would likely sequester the stalk tethered agonist within the interior of the GAIN domain, which would presumably interfere with interactions between stalk tethered agonist residues and the remainder of the CTF and thus loss of the stalk-mediated signaling mechanism. Furthermore, there are 10 single nucleotide polymorphisms reported within the stalk sequence (ADPKD Variant Database; https://pkdb.mayo.edu/welcome), most of which were found to significantly reduce CTF-mediated activation of the NFAT reporter (*Magenheimer et al., 2021*). In particular, the ADPKD-associated G3052R stalk mutation that was previously analyzed along with the stalkless CTF by GaMD simulations (*Pawnikar et al., 2022*) has the same reduction in activity as the stalkless CTF in the cellular signaling reporter assays and the same loss of active/closed conformation interactions in GaMD analyses. Therefore, the stalkless CTF was used in our docking and simulation studies as representative of a biologically relevant mutant form of PC1.

To reveal the molecular mechanisms of the soluble stalk-derived peptides, we chose to perform HPEPDOCK docking and novel Pep-GaMD simulations to sample the peptide interactions with the ΔStalk PC1 CTF. Pep-GaMD simulations were able to refine the docking conformations of peptide agonists bound to the ΔStalk PC1 CTF. It is important to note that the free energy profiles calculated from GaMD simulations of PC1 CTF were not fully converged since certain variations were observed among the individual simulations. Nevertheless, these calculations allowed us to identify representative low-energy binding conformations of the peptides. Pep-GaMD simulations sampled an antiparallel ß-strand and a short helical secondary structure of peptide p17 bound to the ΔStalk CTF. Furthermore, peptides p9 and p21 adopted a more folded structure as compared to their disordered loop conformations in the docking poses. We also observed TOP-PL interactions, particularly the salt bridge between residues R3848-E4078 that is a key feature of the stalk TA-mediated activation of signaling for PC1 CTF (*Pawnikar et al., 2022*). Signal transduction was initiated upon binding of the stalk (TA) to the TOP domain, which was transmitted to the PL via a salt bridge formation between residue R3848 in the TOP domain and residue E4078 in the PL. The bound peptide agonists p9, p17, and p21 maintained the ΔStalk CTF in its 'Closed/Active' conformation as observed in the wild-type PC1 CTF simulations (*Pawnikar et al., 2022*).

The interacting pairs identified using sequence-based covariation analysis matched the pairs identified by Pep-GaMD simulations, providing complementary evidence of the importance of these interactions and of the existence of the 'Bound' and 'Intermediate' binding states of the stalk TA and stalk-derived peptide agonist. This suggests that such stalk TA binding states are evolutionarily conserved across PC1 orthologs. Covariation analysis identifies interactions important during any part of the protein lifecycle, and alone cannot be used to distinguish which conformational state an interaction arises in. By comparison to the conformations found in the Pep-GaMD simulations, we found that most of the identified interactions between the stalk TA and the TOP domain were consistent with either the 'Intermediate' or 'Bound' binding states of the stalk-derived peptides, which are related to CTF inactive and active signaling states, however, it remained possible that other interactions, such as between the start of the stalk-TA and TM2/TM3, may be related to conformational states necessary for cleavage of the GAIN/GPS domain. Additionally, structural contacts may be incompletely detected at some positions when the statistical signal of covariation is masked by high conservation, subfamily specialization, or misalignment. This can explain why some interactions identified in the binding interface through docking are not detected using covariation analysis. Despite this, the specific subset

of interactions detected using covariation analysis suggest broader peptide binding interfaces, and we found these to be consistent with the peptide binding interactions observed in the Pep-GaMD simulations and the MM/PBSA binding free energy analysis, and the covarying residue pairings were consistent with functional biophysical interactions.

The proposed binding interactions of the PC1 stalk peptides share some similarities with those observed for the ADGRs. Specifically, *Xiao et al., 2022* resolved cryo-EM structures of active ADGRG2 and ADGRG4 in complex with tethered Stachel sequences. The structures showed that the 15 residue Stachel sequence inserts into the TM bundle to form intense hydrophobic interactions. A hydrophobic F/Y/LXφφφXφ motif identified in the ADGR tethered sequences formed five finger-like projections in the hydrophobic pits of the TM bundle (*Xiao et al., 2022*). In our study, we observed a similar pattern of intense hydrophobic interactions between the peptide agonists p9, p17, and p21 and the hydrophobic pockets in the TOP domain of PC1 CTF. Notably, a closely similar TOP binding pocket was identified for interaction of the tethered agonist (Stalk) in our previous study (*Pawnikar et al., 2022*) and for binding of peptide agonist p21 in this study. The TOP domain hydrophobic pocket may serve as a significant candidate binding site for designing new synthetic peptides or small molecules to aid in the rescue of PC1 function levels. Moreover, the shorter peptide agonists' (p9 and p17) binding sites also serve as novel pockets for the design and development of therapeutic approaches for treating ADPKD. While the present study is focused on the identification of initial peptides that are able to activate the PC1 CTF, we shall include further mutation experiments and simulations, peptide SAR, and optimization of the lead peptides in future studies. It is also important to note that we have not tested the selectivity of the peptides for PC1 versus PC2 in the present study primarily because transfection of PC2 does not activate the NFAT reporter. However, it is possible that co-transfection of PC2 with the PC1 CTF could alter the stalk peptide binding. This will be important to consider in future studies.

# Materials and methods
## Experimental procedures
### Antibodies and peptide synthesis
Primary antibodies used included A19, for detection of mouse PC1 CTF (*Sutters et al., 2001*), and TRAM2 (Epitomics, 3685–1). Secondary antibodies conjugated to HRP were purchased from Sigma or Jackson ImmunoResearch. Stalk-derived peptides were synthesized by GenScript using the Fmoc method and verified by HPLC-MS analysis.

### DNA expression constructs and cloning
To produce CTF expression construct of mouse (m) PC1, sequences starting at T3041 and proceeding past the first TM domain were amplified by PCR from PC1-11TM (*Puri et al., 2004*), respectively, using 5'-mCleavStalkBsm-For and 3'-TMI-EcoRV primers to produce mCleavStk, which was joined via the BsmBI site to a PCR product encoding the signal peptide sequence (MPMGSLQPLATLYLLG MLVASVLG) from the T cell surface glycoprotein CD5 (*Nims et al., 2003*) in pBlueScript (pBS) to generate pBS-mCD5-cleaved stalk. The EcoRI-EagI fragment containing mCD5-cleaved stalk was joined to a 3.2 kb EagI-NotI fragment from PC1-11TM encoding TM2 through the C-tail to produce the final pCIneo-mCTF expression construct. The stalkless CTF mutant expression construct, pCIneo-mCTF$^{\Delta st}$, starting with S3062 of the mouse PC1 stalk was generated by the same scheme except for using the 5'-mΔStalkBsmFor primer for the initial PCR. PCR and mutagenesis primers were synthesized by IDT and sequences are as follows:

5'-CD5 Eco: 5'- TTCTAGAATTCCCTCGACCTCG –3'; 3'-CD5-BsmBI: 5'- GACTAGCGTCTCATGC CTAGCACGGAAGC –3'; mCleavStalkBsm-For: 5'- GACTAGCGTCTCAGGCACTGCCTTCGGTGCC-3'; mΔStalkBsmFor: 5'- GACTAGCGTCTCAGGCAGTGCAAGCATCAACTACATTGTCC –3'; TMI-EcoRV: 5'-GACTAGGATATCCCTCTGGACTCTAGTAAAGCG-3'; BsrGIstalk-Rev: 5'- AGGGTCTGGGTA GAGTGCTT –3'.

PCR and mutagenesis products and their final constructs were confirmed by Sanger sequencing (GeneWiz). Expression constructs of CTF and CTF$^{\Delta st}$ from human PC1 were made in the pCI vector as described previously (*Pawnikar et al., 2022*). In the conduct of research utilizing recombinant DNA, the investigator adhered to NIH Guidelines for research involving recombinant DNA molecules.

## Cell culture and transient transfection

HEK293T cells (ATCC) were maintained and transiently transfected as described previously (*Maser et al., 2003*). Cells were passaged into 6-well plates (6×10$^5$ cells/well; 3 wells/transfection condition) and transfected with a DNA mixture containing either the 4xNFAT or the 7xAP-1 promoter-Firefly luciferase reporter (100 ng; Stratagene), along with Renilla luciferase (50 ng of pGL4.70[*hRluc*] or 1 ng of pRL-null; Promega), and pCI expression vector encoding either CTF or CTF$^{\Delta st}$ (75 ng for signaling; 600 ng for surface biotinylation) or an equimolar amount of empty pCI vector as control. pBlueScript (Stratagene) was used to bring the total DNA amount to 8 ug. After 2.5 hr, the DNA mixture was replaced with serum-free culture medium, and after 20–24 hr, cells were lysed in 1 X Passive Lysis buffer (PLB; Promega) supplemented with protease and phosphatase inhibitors. Firefly (Fluc)- and Renilla (Rluc)-luciferase activity in each cell lysate was determined using the Dual Luciferase Assay Kit (Promega) and a Berthold tube luminometer. NFAT-Fluc luminescence was normalized to Rluc for each well within a transfection condition, and then averaged for each condition (n=3 wells/condition). Means of normalized NFAT-Fluc with standard deviation (*Dalagiorgou et al., 2013*) were graphed. Signaling-transfection experiments were performed a minimum of three times (i.e., >3 biological replicates) each with three technical replicates/condition except where noted.

## Stalk peptide treatment

Cells were plated into 24-well plates (1.5×10$^5$ cells/well) and transfected with CTF$^{\Delta st}$ or empty pCI expression vectors, along with NFAT-Fluc and Rluc plasmids. Two hours following medium exchange, one-half of the culture medium volume was replaced with an equal volume of either serum-free medium (no peptide control), or stalk-derived or solubility tag peptide (2 mM in serum-free medium) and incubated overnight. In some experiments, an additional 50–100 ul of peptide (1 mM) was added the following morning. Cells were lysed at 24 hr following the initial peptide or control medium addition.

## Cell surface biotinylation analyses

Surface labeling (*Pavel et al., 2014*) was performed on intact cells 22–24 hr post-transfection using 1.5 mg/ml PBS of the membrane-impermeable, cleavable biotin cross-linking reagent (Sulfo-NHS-SS-Biotin; Pierce) for 30 min on ice. Crosslinking was inactivated by the addition of 50 mM Tris, pH 8.0. Cells were washed and then lysed in 1 X PLB with protease inhibitors. A 10% aliquot of the total cell lysate was removed and saved as the input sample. NeutraAvidin-agarose beads (Pierce) were added to remove biotinylated surface proteins. The supernatant was removed and saved as the unbound cytosolic fraction (sup). A representative amount of each fraction, i.e., the total lysate (input), cytosolic (sup), and biotinylated surface proteins (beads) was analyzed by SDS-PAGE/Western blot. TRAM2 (ER-resident protein) was used as a cell fractionation marker.

## Western blot analyses

Gel loading volumes of signaling cell lysates were calculated based on normalization to relative Rluc activity within a condition. Lysate samples were electrophoresed through 7.5% denaturing polyacrylamide gels and transferred to nitrocellulose membrane using the TransBlot Turbo and ReadyBlot transfer buffer (BioRad). Blots were incubated with primary antibody (1:1000 dilution for A19; 1:10,000 for anti-TRAM2) in Tris-buffered saline (TBS; 10 mM Tris pH 7.4, 150 mM NaCl) with 0.1% Tween-20 (TBST) and 5% powdered dry milk for 14–16 hr at 4 °C. Secondary antibodies conjugated to HRP were incubated at 1:5,000 dilution in TBST/5% milk for 1 hr at room temperature. Blots were developed using a chemiluminescent substrate (Clarity; BioRad), and multiple exposures were captured for each blot with the Amersham Imager 600 with the band saturation detection mode enabled. Volume density (minus background) of immunoreactive bands was determined using ImageQuant TL (GE Healthcare). In most cases, duplicate blots were prepared and quantified to obtain an average band density relative to wild-type CTF.

## Statistical analyses

Statistical analysis was performed using GraphPad Prism 9 (GraphPad Software, San Diego, CA, USA). Data are presented as means + standard deviation (*Dalagiorgou et al., 2013*) for bar graphs. Multiple

comparisons used one-way ANOVA and Tukey's post-hoc analysis. $p \leq 0.05$ was considered statistically significant.

## Computational Methods

### Gaussian accelerated molecular dynamics (GaMD)

GaMD is an unconstrained enhanced sampling approach that works by adding a harmonic boost potential to smooth the potential energy surface of biomolecules to reduce energy barriers (*Miao et al., 2015*). Brief description of the method is provided here.

Consider a system with $N$ atoms at positions $\vec{r} = \{\vec{r}_1, \ldots, \vec{r}_N\}$. When potential energy of the system $V(\vec{r})$ is less than a threshold energy $E$, a boost potential $\Delta V(\vec{r})$ is added to the system as follows:

$$V^*(\vec{r}) = V(\vec{r}) + \Delta V(\vec{r}), \, V(\vec{r}) < E \tag{1}$$

$$\Delta V(\vec{r}) = \frac{1}{2} k \left( E - V(\vec{r}) \right)^2, \, V(\vec{r}) < E, \tag{2}$$

where $k$ is the harmonic force constant. The two adjustable parameters $E$ and $k$ can be determined by the application of three enhanced sampling principles. First, for any two arbitrary potential values $V_1(\vec{r})$ and $V_2(\vec{r})$ found on the original energy surface, if $V_1(\vec{r}) < V_2(\vec{r})$, $\Delta V$ should be a monotonic function that does not change the relative order of the biased potential values, i.e., $V_1^*(\vec{r}) < V_2^*(\vec{r})$. Second, if $V_1(\vec{r}) < V_2(\vec{r})$, the potential difference observed on the smoothed energy surface should be smaller than that of the original, i.e., $V_2^*(\vec{r}) - V_1^*(\vec{r}) < V_2(\vec{r}) - V_1(\vec{r})$. By combining the first two criteria and plugging in the formula of $V^*(\vec{r})$ and $\Delta V$, we obtain:

$$V_{max} \leq E \leq V_{min} + \frac{1}{k}, \tag{3}$$

where $V_{min}$ and $V_{max}$ are the system's minimum and maximum potential energies. To ensure that *Equation 3* is valid, $k$ has to satisfy: $k \leq \frac{1}{V_{max} - V_{min}}$. Let us define $k \equiv \frac{k_0}{V_{max} - V_{min}}$, then $0 < k_0 \leq 1$. Third, the standard deviation (SD) of $\Delta V$ needs to be small enough (i.e., narrow distribution) to ensure accurate reweighting using cumulant expansion to the second order: $\sigma_{\Delta V} = k \left( E - V_{avg} \right) \sigma_V \leq \sigma_0$, where $V_{avg}$ and $\sigma_V$ are the average and SD of $\Delta V$ with $\sigma_0$ as a user-specified upper limit (e.g. $10 k_B T$) for accurate reweighting. When $E$ is set to the lower bound $E = V_{max}$ according to *Equation 3*, $k_0$ can be calculated as:

$$k_0 = \min\left( 1.0, k_0' \right) = \min\left( 1.0, \frac{\sigma_0}{\sigma_V} \cdot \frac{V_{max} - V_{min}}{V_{max} - V_{avg}} \right) \tag{4}$$

Alternatively, when the threshold energy $E$ is set to its upper bound $E = V_{min} + \frac{1}{k}$, $k_0$ is set to:

$$k_0 = k_0'' \equiv \left( 1 - \frac{\sigma_0}{\sigma_V} \right) \cdot \frac{V_{max} - V_{min}}{V_{avg} - V_{min}} \tag{5}$$

if $k_0''$ is calculated between 0 and 1. Otherwise, $k_0$ is calculated using *Equation (4)*.

### Peptide Gaussian accelerated molecular dynamics (Pep-GaMD)

Peptides often undergo large conformational changes during binding to target proteins, being distinct from small-molecule ligand binding or protein-protein interactions (PPIs). In this regard, Peptide GaMD or 'Pep-GaMD' has been developed to enhance the sampling of peptide binding (*Wang and Miao, 2020*). In Pep-GaMD, we consider a system of peptide $L$ binding to a protein $P$ in a biological environment $E$. Presumably, peptide binding mainly involves in both the bonded and non-bonded interaction energies of the peptide since peptides often undergo large conformational changes during binding to the target proteins. Thus, the essential peptide potential energy is $V_L(r) = V_{LL,b}(r_L) + V_{LL,nb}(r_L) + V_{PL,nb}(r_{PL}) + V_{LE,nb}(r_{LE})$. In Pep-GaMD, we add boost potential selectively to the essential peptide potential energy according to the GaMD algorithm:

$$\Delta V_L\left(r\right) = \begin{cases} \frac{1}{2}k_L\left(E_L - V_L\left(r\right)\right)^2, & V_L\left(r\right) < E_L \\ 0, & V_L\left(r\right) \geq E_L \end{cases} \tag{6}$$

where $E_L$ is the threshold energy for applying boost potential and $k_L$ is the harmonic constant. In addition to selectively boosting the peptide, another boost potential is applied to the protein and solvent to enhance conformational sampling of the protein and facilitate peptide rebinding. This boost represents the total system potential energy without the essential peptide potential energy included:

$$\Delta V_D\left(r\right) = \begin{cases} \frac{1}{2}k_D\left(E_D - V_D\left(r\right)\right)^2, & V_D\left(r\right) < E_D \\ 0, & V_D\left(r\right) \geq E_D \end{cases} \tag{7}$$

Where $V_D$ represents the total system potential energy without the essential peptide potential energy included, $E_D$ represents the second boost potential threshold energy and $k_D$ represents the harmonic constant. Hence, this contributes to the dual-boost Pep-GaMD as the total boost potential $\Delta V\left(r\right) = \Delta V_L\left(r\right) + \Delta V_D\left(r\right)$.

## Energetic reweighting of Pep-GaMD simulations

For energetic reweighting of Pep-GaMD simulations to calculate potential mean force (PMF), the probability distribution along a reaction coordinate is written as $p^*\left(A\right)$. Given the boost potential $\Delta V\left(r\right)$ of each frame, $p^*\left(A\right)$ can be reweighted to recover the canonical ensemble distribution $p\left(A\right)$, as:

$$p\left(A_j\right) = p^*\left(A_j\right)\frac{\left\langle e^{\beta\Delta V\left(r\right)}\right\rangle_j}{\sum_{i=1}^{M}\left\langle p^*\left(A_i\right)e^{\beta\Delta V\left(r\right)}\right\rangle_i}, j = 1,, M \tag{8}$$

where $M$ is the number of bins, $\beta = k_B T$ and $\left\langle e^{\beta\Delta V\left(r\right)}\right\rangle_j$ is the ensemble-averaged Boltzmann factor of $\Delta V\left(r\right)$ for simulation frames found in the $j^{\text{th}}$ bin. The ensemble-averaged reweighting factor can be approximated using cumulant expansion:

$$\left\langle e^{\beta\Delta V\left(r\right)}\right\rangle_j = \exp\left\{\sum_{k=1}^{\infty}\frac{\beta^k}{k!}C_k\right\} \tag{9}$$

where the first two cumulants are given by

$$C_1 = \langle\Delta V\rangle$$
$$C_2 = \left\langle\Delta V^2\right\rangle - \langle\Delta V\rangle^2 = \sigma_V^2 \tag{10}$$

The boost potential obtained from Pep-GaMD simulations usually follows near-Gaussian distribution. Cumulant expansion to the second order thus provides a good approximation for computing the reweighting factor. The reweighted free energy $F\left(A\right) = -k_B T\ln p\left(A\right)$ is calculated as

$$F\left(A\right) = F^*\left(A\right) - \sum_{k=1}^{2}\frac{\beta^k}{k!}C_k + F_c \tag{11}$$

where $F^*\left(A\right) = -k_B T\ln p^*\left(A\right)$ is the modified free energy obtained from GaMD simulation and $F_c$ is a constant.

## Computational model of peptide agonist-bound ΔStalk PC1 CTF and HPEP-DOCK docking

With GaMD simulations of the WT PC1 CTF obtained from the previous study that revealed an active TA/stalk-mediated allosteric signaling (*Pawnikar et al., 2022*), structural clustering of the extracellular regions of PC1 CTF, including the Stalk, TM2-TM3, and TM4-TM5 loops, TOP domain, S3-S4 loop and pore loop, was performed using the hierarchical agglomerative algorithm in CPPTRAJ (*Roe and

*Cheatham, 2013*). The top-ranked representative conformation of PC1 CTF was used for peptide docking after removal of the TA/stalk (ΔStalk). Then the HPEPDOCK (*Zhou et al., 2018*) webserver was applied to dock the p9, p17, and p21 stalk-derived peptides to ΔStalk CTF.

## Simulation system setup

We embedded ΔStalk CTF in a palmitoyl-oleoyl-phosphatidyl-choline (POPC) bilayer and solvated the system in 0.15 M NaCl explicit solvent using CHARMM-GUI (*Figure 2—figure supplement 1C*). Neutral patches (acetyl and methylamide) were added to the protein termini residues. The peptide termini were kept as charged (NH3 + and COO-). The CHARMM36m (*Vanommeslaeghe and MacKerell, 2015*) force field parameters were used for the protein, peptides, and lipids. CHARMM-GUI output files and scripts were used with default parameters to prepare the systems for Pep-GaMD simulations. Energy minimization was performed for 5000 steps using a constant number, volume, and temperature (NVT) ensemble at 310 K. Further equilibration was done for 375 ps at 310 K using an NPT ensemble. Conventional MD (cMD) simulations was performed on the systems for 10 ns at 1 atm pressure and 310 K temperature. All-atom Pep-GaMD simulations were performed with a short cMD for 10 ns, Pep-GaMD equilibration for 55 ns followed by three independent Pep-GaMD production runs for 500 ns for each system with randomized initial atomic velocities. A cutoff distance of 9 Å was used for the van der Waals and short-range electrostatic interactions, and long-range electrostatic interactions were computed with the particle-mesh Ewald summation method (*Darden et al., 1993*). The simulation systems were ~90×136×117 Å$^3$ in dimension, containing a total of ~100 K atoms with explicit solvent and lipid molecules.

## Simulation analysis

Pep-GaMD simulation trajectories were analyzed using CPPTRAJ (*Roe and Cheatham, 2013*) and VMD (*Humphrey et al., 1996*) tools. Trajectory analysis showed peptides binding to the TOP domain of PC1 CTF. A previously identified salt bridge formed between the TOP domain R3848 and the pore loop E4078 of the PC1 protein is an important interaction during PC1 signal activation. The number of contacts formed between peptides p9, p17 and p21, respectively, and the R3848-E4078 salt bridge distance were used as reaction coordinates to calculate 2D free energy profiles using the *PyReweighting* toolkit (*Miao et al., 2014*). A bin size of 2 Å was used for distances and 10 for the number of contacts. Three independent Pep-GaMD simulations were combined to perform structural clustering using the hierarchical agglomerative clustering algorithm in CPPTRAJ (*Roe and Cheatham, 2013*). A 3 Å RMSD cutoff was used for each peptide system. PyReweighting (*Miao et al., 2014*) was then applied to calculate the original free energy values of each peptide structural cluster with a cutoff of 500 frames. The structural clusters were finally ranked according to the reweighted free energy values.

## Potts sequence covariation model

Using a seed alignment of 189 orthologs of human *PKD1* obtained from the Ensemble database (*Aoto et al., 2016*), we used iterative searching of the UniProt Database using HHblits (*Howe et al., 2021*) to obtain a multiple-sequence alignment of 4384 homologs, and after filtering using standard methods with an 80% identity threshold (*Haldane and Levy, 2021*), we obtained 1022 effective sequences of length 853. These sequences had an average of 23% sequence identity reflecting extensive diversity across Eukaryotes. As this number of sequences and sequence length could lead to an overfit model (*Haldane and Levy, 2019*), we used a subset the MSA to limit to 394 positions on the extracellular side of *PKD1* including the GAIN domain, stalk, the TOP domain, and ends of the transmembrane helices. We inferred a Potts model from this reduced MSA using the Mi3-GPU software (*Haldane and Levy, 2021*). We evaluated the position-pair statistical interactions using the 'weighted Frobenius norm' interaction score (*Haldane and Levy, 2021*), and computed the residue-residue covariation values $C_{ab}^{ij} = f_{ab}^{ij} - f_a^i f_b^j$ as the difference between the pair-residue frequency $f_{ab}^{ij}$ of letters a and b at positions i,j and the null expectation under the assumption of site-independence by multiplying the two single-site frequencies, $f_a^i$ and $f_b^j$. The maximum possible covariance is 25%. Statistically significant covariations scores will be greater than the expected binomial sampling error given our dataset size of N~1022,, and for bivariate count $f_{ab}^{ij}$ the binomial-sampling standard deviation is $\sqrt{f_{ab}^{ij}\left(1 - f_{ab}^{ij}\right)/N}$

which for the typical bivariate frequency of ~10% corresponds to a ~1% error. To choose a cutoff in Frobenius Norm to distinguish likely contacts from noise, we compared the Potts interactions scores to the contacts predicted using the Alphafold (*Jumper et al., 2021*) structure, choosing the plotting cutoff at a false-positive rate of 50% relative to the contacts predicted using the Alphafold structure using a 8 Å nearest heavy atom side chain distance.

## Peptide binding free energy calculations

Molecular Mechanics/Poisson-Boltzmann Surface Area (MM/PBSA) analysis was performed to calculate the binding free energies of peptides p9, p17, and p21 to PC1 CTF. The analysis was performed using the trajectory in which the peptide was bound to the receptor. In MM/PBSA (*Wang et al., 2019*), the binding free energy of the ligand (L) to the receptor (R) to form the complex (RL) is calculated as:

$$\Delta G_{bind} = G_{RL} - G_R - G_L \tag{12}$$

where $G_{RL}$ is the Gibbs free energy of the complex RL, $G_R$ is the Gibbs free energy of the molecule R in its unbound state and $G_L$ is the Gibbs free energy of the molecule L in its unbound state, respectively. $\Delta G_{bind}$ can be divided into contributions of different interactions as (*Srinivasan et al., 1998*):

$$\Delta G_{bind} = \Delta H - T\Delta S = \Delta E_{MM} + \Delta G_{sol} - T\Delta S \tag{13}$$

in which

$$\Delta E_{MM} = \Delta E_{int} + \Delta E_{elec} + \Delta E_{vdW} \tag{14}$$
$$\Delta G_{sol} = \Delta G_{PB/GB} + \Delta G_{SA} \tag{15}$$
$$\Delta G_{SA} = \gamma.SASA + b \tag{16}$$

where $\Delta E_{MM}$, $\Delta G_{sol}$, $\Delta H$, and $-T\Delta S$ are the changes in the gas-phase molecular mechanics (MM) energy, solvation-free energy, enthalpy, and conformational entropy upon ligand binding, respectively. $\Delta E_{MM}$ includes the changes in the internal energies $\Delta E_{int}$ (bond, angle, and dihedral energies), electrostatic energies $\Delta E_{elec}$, and the van der Waals energies $\Delta E_{vdW}$. $\Delta G_{sol}$ is the sum of the electrostatic solvation energy $\Delta G_{PB/GB}$ (polar contribution) and the nonpolar contribution $\Delta G_{SA}$ between the solute and the continuum solvent. The polar contribution is calculated using either the Poisson Boltzmann (PB) or Generalized Born (GB) model, while the nonpolar energy is usually estimated using the solvent-accessible surface area (SASA) (*Gilson and Honig, 1988*; *Wang et al., 2006*) where γ is surface tension coefficient and b is the constant offset. The change in conformational entropy $-T\Delta S$ is usually calculated by normal-mode analysis (*Srinivasan et al., 1998*) on a set of conformational snapshots taken from MD simulations. However, due to the large computational cost, changes in the conformational entropy are usually neglected as we were concerned more on relative binding free energies of the similar peptide ligands.

MM/PBSA analysis was performed using the gmx_MMPBSA (*Valdés-Tresanco et al., 2021*) software with the following command line:

```
gmx_MMPBSA -O -i mmpbsa.in -cs com.tpr -ci index.ndx -cg 1 13 -ct com_traj.
xtc -cp topol.top -o FINAL_RESULTS_MMPBSA.dat -eo FINAL_RESULTS_MMPBSA.csv
```

Input file for running MM/PBSA analysis:

```
&general
sys_name="Prot-Pep-CHARMM",
startframe=1,
endframe=200,
# In gmx_MMPBSA v1.5.0 we have added a new PB radii set named charmm_radii
# This radii set should be used only with systems prepared with CHARMM
force fields.
# Uncomment the line below to use charmm_radii set
#PBRadii=7,
/
&pb
# radiopt=0 is recommended which means using radii from the prmtop file for
```

```
both the PB calculation and for the NP
# calculation
istrng=0.15, fillratio=4.0, radiopt=0
```

Residue-wise interaction energy analysis was performed on peptides p9, p17, and p21 using the trajectory in which the peptide was bound to the PC1 CTF using the gmx_MMPBSA (*Valdés-Tresanco et al., 2021*) software with the following command line:

```
gmx_MMPBSA -O -i mmpbsa.in -cs com.tpr -ct com_traj.xtc -ci index.ndx -cg
3 4 -cp topol.top -o FINAL_RESULTS_MMPBSA.dat -eo FINAL_RESULTS_MMPBSA.csv
-do FINAL_DECOMP_MMPBSA.dat -deo FINAL_DECOMP_MMPBSA.csv
```

Input file for running residue-wise energy decomposition analysis:

```
&general
sys_name="Decomposition",
startframe=1,
endframe=200,
#forcefields="leaprc.protein.ff14SB"
/
&gb
igb=5, saltcon=0.150,
/
#make sure to include at least one residue from both the receptor
#and peptide in the print_res mask of the &decomp section.
#this requirement is automatically fulfilled when using the within keyword.
#http://archive.ambermd.org/201308/0075.html
&decomp
idecomp=2, dec_verbose=3,
print_res="A/854-862 A/1-853",
/
```

## Acknowledgements

We thank Dr. Yan Zhang for valuable discussions. This work used supercomputing resources with allocation award TG-MCB180049 through the Advanced Cyberinfrastructure Coordination Ecosystem: Services & Support (ACCESS) program, which is supported by National Science Foundation grants #2138259, #2138286, #2138307, #2137603, and #2138296, and project M2874 through the National Energy Research Scientific Computing Center (NERSC), which is a U.S. Department of Energy Office of Science User Facility operated under Contract No. DE-AC02-05CH11231, and the Research Computing Cluster at the University of Kansas. This work was supported in part by National Institutes of Health (R01DK123590 and R56DK135824), Department of Defense CDMRP PRMRP Discovery Award (PR160710/W81XWH-17-1-0301), and Pilot Grant funding from the School of Health Professions at KU Medical Center (to RLM), Pilot award 1004015 from Jared Grantham Kidney Institute at KU Medical Center (to YM and RLM), the National Institutes of Health (R56DK135824) and startup project 27110 at University of North Carolina - Chapel Hill (to YM), and the National Institutes of Health (R35-GM132090 and OD020095) (to AH). This research includes calculations carried out on HPC resources supported in part by the National Science Foundation through major research instrumentation grant number 1625061 and by the US Army Research Laboratory under contract number W911NF-16-2-0189.

# Additional information

## Funding

| Funder | Grant reference number | Author |
|---|---|---|
| National Institutes of Health | R56DK135824 | Allan Haldane<br>Robin L Maser<br>Yinglong Miao |
| National Energy Research Scientific Computing Center | Project M2874 | Yinglong Miao |
| National Institutes of Health | R01DK123590 | Robin L Maser |
| Department of Defense Education Activity | CDMRP PRMRP Discovery Award (PR160710/ W81XWH-17-1-0301) | Robin L Maser |
| Jared Grantham Kidney Institute, KU Medical Center | Pilot award 1004015 | Robin L Maser |
| University of North Carolina - Chapel Hill | Startup project 27110 | Yinglong Miao |
| National Institutes of Health | R35-GM132090 and OD020095 | Allan Haldane |
| National Science Foundation | MRI1625061 | Allan Haldane |
| US Army Research Laboratory | W911NF-16-2-0189 | Allan Haldane |

The funders had no role in study design, data collection and interpretation, or the decision to submit the work for publication.

## Author contributions

Shristi Pawnikar, Data curation, Formal analysis, Investigation, Visualization, Methodology, Writing – original draft; Brenda S Magenheimer, Data curation, Formal analysis, Validation, Investigation, Visualization, Methodology; Keya Joshi, Data curation, Formal analysis, Investigation, Visualization, Writing – review and editing; Ericka Nevarez-Munoz, Formal analysis, Validation, Investigation, Visualization; Allan Haldane, Data curation, Formal analysis, Validation, Investigation, Visualization, Methodology, Writing – original draft, Writing – review and editing; Robin L Maser, Conceptualization, Formal analysis, Writing – original draft, Writing – review and editing; Yinglong Miao, Conceptualization, Formal analysis, Supervision, Validation, Investigation, Writing – original draft, Writing – review and editing

## Author ORCIDs

Keya Joshi ⬛ https://orcid.org/0009-0001-8139-478X
Allan Haldane ⬛ https://orcid.org/0000-0002-8343-1994
Yinglong Miao ⬛ https://orcid.org/0000-0003-3714-1395

Reviewer #1 (Public Review): https://doi.org/10.7554/eLife.95992.3.sa1
Reviewer #2 (Public Review): https://doi.org/10.7554/eLife.95992.3.sa2
Reviewer #3 (Public Review): https://doi.org/10.7554/eLife.95992.3.sa3
Author response https://doi.org/10.7554/eLife.95992.3.sa4

# Additional files

## Supplementary files
• MDAR checklist

## Data availability

The authors declare that the data supporting the findings of this study are available within the paper. files have been uploaded in Figshare database with DOIs https://doi.org/10.6084/m9.figshare.26793067.v1; https://doi.org/10.6084/m9.figshare.26792983.v1; https://doi.org/10.6084/m9.figshare.26793088.v1.

The following datasets were generated:

| Author(s) | Year | Dataset title | Dataset URL | Database and Identifier |
|---|---|---|---|---|
| Pawnikar S, Magenheimer BS, Joshi K, Munoz EN, Haldane A, Maser RL, Miao Y | 2024 | p9 peptide simulation data. Dataset includes imaged trajectory files for the five Pep-GaMD simulations runs. We have also provided the topology and the pdb file | https://doi.org/10.6084/m9.figshare.26793067.v1 | figshare, 10.6084/m9.figshare.26793067.v1 |
| Pawnikar S, Magenheimer BS, Joshi K, Munoz EN, Haldane A, Maser RL, Miao Y | 2024 | p17 peptide simulation data. Dataset includes imaged trajectory files for the five Pep-GaMD simulations runs. We have also provided the topology and the pdb file | https://doi.org/10.6084/m9.figshare.26792983.v1 | figshare, 10.6084/m9.figshare.26792983.v1 |
| Pawnikar S, Magenheimer BS, Joshi K, Munoz EN, Haldane A, Maser RL, Miao Y | 2024 | p21 peptide simulation data. Dataset includes imaged trajectory files for the five Pep-GaMD simulations runs. We have also provided the topology and the pdb file | https://doi.org/10.6084/m9.figshare.26793088.v1 | figshare, 10.6084/m9.figshare.26793088.v1 |

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
