## [Editor Report · eLife assessment]

This joint computational/experimental study demonstrates the ability of synthetic peptides derived from the stalk tethered agonist in polycystin-1 (PC1) to re-activate signaling by a stalkless C-terminal fragment of PC1. The study is **valuable** as it discovered peptide agonists for PC1 and the integrated *in vitro* and *in silico* approach is potentially applicable to the analysis of related systems. Following the revision, the line of evidence presented in the current manuscript is considered **convincing**.

---

## [Referee Report · Reviewer #1 (Public Review)]

Summary:

This research used cell-based signaling assay and Gaussian-accelerated molecular dynamics (GaMD) to study peptide-mediated signaling activation of polycystin-1 (PC1), which is responsible for the majority of autosomal dominant Polycystic Kidney Disease (ADPKD) cases. Synthetic peptides of various lengths derived from the N-terminal portion of the PC1 C-terminal fragment (CTF) were applied to HEK293T cells transfected with stalkless mouse CTF expression construct. It was shown that peptides including the first 7, 9, and 17 residues of the N-terminal portion could activate signaling to the NFAT reporter. To further understand the underlying mechanism, docking and peptide-GaMD simulations of peptides composed of the first 9, 17, and 21 residues from the N-terminal portion of the human PC1 CTF were performed. These simulations revealed the correlation between peptide-CTF binding and PC1 CTF activation characterized by the close contact (salt bridge interaction) between residues R3848 and E4078. Finally, a Potts statistical model was inferred from diverged PC1 homologs to identify strong/conserved interacting pairs within PC1 CTF, some of which are highly relevant to the findings from the peptide GaMD simulations. The peptide binding pockets identified in the GaMD simulations may serve as novel targets for design of therapeutic approaches for treating ADPKD.

Strengths:

(1) The experimental and computational parts of this study complement and mostly support each other, thus increasing the overall confidence in the claims made by the authors.

(2) The use of exogenous peptides and a stalkless CTF in the GaMD is a step forward compared to earlier simulations using the full CTF, CTF mutants, or the stalkless CTF alone. And it led to findings of novel binding pockets.

(3) Since the PC1 shares characteristics with the Adhesion class of GPCRs, the approaches used in this work may be extended to other similar systems.

Weaknesses:

(1) Only results for selective peptides (p9, p17 p21) binding with the protein were shown. It would be interesting to see the interaction between some (if not all) of the other peptides with the protein.

(2) The convergence of the simulations is not very good. The results should be interpreted more qualitatively rather than quantitively because large variations in the free energy profile were seen between different replicates. Although these simulations might have identified representative low-energy binding conformations of the peptides, whether they have explored all possible conformations is still a question.

---

## [Referee Report · Reviewer #2 (Public Review)]

Summary:

This manuscript, "Activation of polycystin-1 Signaling by Binding of Stalk-derived Peptide Agonists", by Miao and coworkers. The autosomal dominant Polycystic Kidney Disease (ADPKD) is a major form of Polycystic Kidney Disease (PKD). To provide better treatment and avoid side effects associated with currently available options, the authors investigated an interesting GPCR, polycystin-1 (PC1), as a potential therapeutic target. *In vitro* and *in silico* studies were combined to identify peptide agonists for PC1 and to elucidate their roles in PC1 signaling. Overall, regarding the significance of the findings, this work described valuable peptide agonists for PC1 and the combined *in vitro* and *in silico* approach can be useful to study a complex system like PC1. However, the strength of the evidence is incomplete, as more experiments are needed as controls to validate the computational observations. The work appears premature.

Strengths:

(1) This work first described the experimental discovery of short peptides designed to mimic the stalk region of PC1, followed by computational investigation using docking and MD simulations. PC1 is a complex membrane protein and an emerging target for ADPKD, but it can be challenging to study. The knowledge and the peptide discovery can be valuable and useful to understand the mechanism and potential modulation of PC1.

(2) The authors published the mechanistic study of PC1 and identified key interacting residues such as N3074-S3585 and R3848-E4078, using very similar techniques (PNAS 2022, 119(19), e2113786119). This work furthers this research by identifying peptides that are stalk mimics for PC1 activation.

(3) Eight peptides were designed and tested experimentally first; three were computationally studied with docking and GaMD simulations to understand their mechanism (s).

Weaknesses:

(1) The selectivity of the peptides between PC1 and PC2 remains unknown in this revision.

Overall, my comments were mostly addressed properly.

---

## [Referee Report · Reviewer #3 (Public Review)]

Summary:

The authors demonstrate the activation of polycystin-1 (PC1), a G-protein coupled receptor, using small peptides derived from its original agonist, the stalk TA protein. In the experimental part of the study, the authors performed cellular assays to check the peptide-induced reactivation of a mutant form of PC1 which does not contain the stalk agonist. The experimental data is supported by computational studies using state-of-the-art Gaussian accelerated Molecular Dynamics (GaMD) and bioinformatics analysis based on sequence covariance. The computer simulations revealed the mechanistic details of the binding of the said peptides with the mutant PC1 protein and discovered different bound, unbound, and intermediate conformations depending on the peptide size and sequence. Due to the use of reliable and well-established molecular simulation algorithms and the physiological relevance of this protein autosomal dominant Polycystic Kidney Disease (ADPKD) make this work particularly valuable.

Strengths:

This work is exploratory and its goal is to establish that small peptides can be used to probe the PC1 signaling process. The authors have provided sufficient evidence to justify this claim. Their GaMD simulations have produced free-energy landscapes that differentiate the interaction of PC1 with three different synthetic peptides and demonstrate the associated conformational dynamics of the receptor protein. Their trajectory analysis and sequence covariance analysis could identify residue-specific interactions that facilitate this process. The authors also performed residue-wise and total interaction energy calculations to substantiate their findings.

Weaknesses:

The reported free energy landscapes are not fully converged. But they are still sufficient to gain biological insight.

---

## [Author Response]

The following is the authors’ response to the original reviews.

**Reviewer 1:**
This research used cell-based signaling assay and Gaussian-accelerated molecular dynamics (GaMD) to study peptide-mediated signaling activation of polycystin-1 (PC1), which is responsible for the majority of autosomal dominant Polycystic Kidney Disease (ADPKD) cases. Synthetic peptides of various lengths derived from the N-terminal portion of the PC1 C-terminal fragment (CTF) were applied to HEK293T cells transfected with stalkless mouse CTF expression construct. It was shown that peptides including the first 7, 9, and 17 residues of the N-terminal portion could activate signaling to the NFAT reporter. To further understand the underlying mechanism, docking and peptide-GaMD simulations of peptides composed of the first 9, 17, and 21 residues from the N-terminal portion of the human PC1 CTF were performed. These simulations revealed the correlation between peptide-CTF binding and PC1 CTF activation characterized by the close contact (salt bridge interaction) between residues R3848 and E4078. Finally, a Potts statistical model was inferred from diverged PC1 homologs to identify strong/conserved interacting pairs within PC1 CTF, some of which are highly relevant to the findings from the peptide GaMD simulations. The peptide binding pockets identified in the GaMD simulations may serve as novel targets for the design of therapeutic approaches for treating ADPKD.

We greatly appreciate the reviewer’s encouraging and positive comments. The reviewer’ specific comments are addressed pointwise below and changes to the text will be highlighted in yellow in the revised manuscript.

(1) The GaMD simulations all include exogenous peptides, thus lacking a control where no such peptide is present (and only stalkless CTF). An earlier study (PNAS 2022 Vol. 119 No. 19 e2113786119) covered this already, but it should be mentioned here that there was no observation of close/activation for the stalkless CTF.

We appreciate the reviewer’s concern about the lack of a control where no exogenous peptide is present. As suggested by the reviewer, we are adding more details about the study on the stalkless CTF as a control in the Introduction of the revised manuscript.

(2) Although 5 independent trajectories were generated for each peptide, the authors did not provide sufficient details regarding the convergence of the simulation. This leaves some uncertainties in their results. Given that the binding poses changed relative to the starting docked poses for all three peptides, it is possible that some other binding pockets and/or poses were not explored.

We appreciate the reviewer’s comment regarding the convergence of the simulation results. This is clarified in the revised manuscript as:

“We have calculated free energy profiles of individual simulations for each system, including the p9, p17, and p21, as shown below (Figs. S5, S6 and S8). For the p9 peptide, the “Bound” lowenergy state was consistently identified in the 2D free energy profile of each individual simulation (Fig. S5). For the p17 peptide, Pep-GaMD simulations were able to refine the peptide conformation from the "Unbound” to the "Intermediate” and “Bound” states in Sim1 and Sim5, while the peptide reached only the "Intermediate” state in the other three simulations (Fig. S6). For the p21 peptide, Pep-GaMD was able to refine the peptide docking conformation to the

"Bound” state in all the five individual simulations (Fig. S8).”

“It is important to note that the free energy profiles calculated from GaMD simulations of PC1 CTF were not fully converged since certain variations were observed among the individual simulations. Nevertheless, these calculations allowed us to identify representative low-energy binding conformations of the peptides.”

(3) The free energy profiles (Figures 2 to 4) based on the selected coordinates provide important information regarding binding and CTF conformational change. However, it is a coarsegrained representation and complementary analysis such as RDFs, and/or contact maps between the peptide and CTF residues might be helpful to understand the details of their interactions. These details are currently only available in the text.

Following the reviewer's suggestion, we have now included a set of protein contact maps showing contacts between the peptides and the TOP domain for each peptide in the representative "Bound” state in revised Supplementary Information (Fig. S4). The contact maps serve to visualize the list of contacts mentioned in the main text. This will be clarified in the revised manuscript.

(4) The use of a stalkless CTF is necessary for studying the functions of the exogenous peptides. However, the biological relevance of the stalkless CTF to ADPKD was not clearly explained, if any.

We appreciate the reviewer’s comment. As correctly assessed by the reviewer, the stalkless CTF is not a biological form of PC1 observed in ADPKD, but rather was used as the simplest or least complex system in which the activities and binding of exogenous peptides could be studied. However, in ADPKD, there are numerous missense mutations reported within the GPCR autoproteolysis-inducing (GAIN) domain that have been shown to prevent or inhibit cleavage at the GPCR-coupled proteolysis site (GPS). Loss of PC1 GPS cleavage, which is known to cause ADPKD, would retain or sequester the stalk tethered agonist within the interior of the GAIN domain, which would presumably interfere with interactions between stalk tethered agonist residues and the remainder of the CTF. Furthermore, there are 10 single nucleotide polymorphisms reported within the stalk sequence (ADPKD Variant Database; https://pkdb.mayo.edu/welcome), most of which we have found to significantly reduce CTF-mediated activation of the NFAT reporter (Magenheimer BS, et al., Constitutive signaling by the C-terminal fragment of polycystin1 is mediated by a tethered peptide agonist; bioRxiv 2021.08.05.455255). In particular, the ADPKD-associated G3052R stalk mutation that was analyzed along with the stalkless CTF by GaMD simulations (Pawnikar et al, PNAS, 2022) has the same reduction in activity as the stalkless CTF in the cellular signaling reporter assays and the same loss of closed conformation interactions in GaMD analyses. As such, we believe the stalkless CTF has biological relevance from the aspect that it mimics the deficiency in signaling activation observed for PC1 CTF stalk mutants. This is clarified in the revised manuscript in the Introduction, page 5, “constructs encoding a stalkless PC1 CTF (a nonbiological mutant of PC1 with deletion of the first 21 N-terminal residues of CTF and three ADPKD-associated…”) and near the beginning of the Discussion, page 16, where the biological relevance of studying the stalkless CTF is explained

(5) The authors might want to clarify if a stalkless CTF is commonly seen in ADPKD, or if it is just a construct used for this study.

The stalkless CTF is not a biological form of PC1, but rather a construct used for this study. This was clarified in the revised manuscript (see response above).

(6) (Pages 7-8) "...we generated expression constructs of mouse (m) PC1 consisting of the CD5 signal peptide sequence fused in frame with the stalk sequence of mCTF ...". What is the CD5 signal peptide sequence here? What is its use?

The CD5 signal peptide sequence is “MPMGSLQPLATLYLLGMLVASVLG” from the T cell surface glycoprotein, CD5. Since the N-terminus of PC1 CTF is derived from a posttranslational, autocatalytic, endoproteolytic cleavage event, this isoform is already membraneembedded and therefore lacks its endogenous signal peptide. The CD5 signal peptide coding sequence is added to the PC1 CTF expression constructs in order to ensure translation and insertion of the encoded protein at the endoplasmic reticulum. Additional details were added to the Experimental Procedures, page 2 of Supporting Information.

(7) (Page 8) "All peptides were appended with a C-terminal, 7-residue hydrophilic sequence (GGKKKKK) to increase solubility". How did the authors make sure that this sequence has no influence on the signaling?

To determine the possible effect of the hydrophilic GGKKKKK sequence on signaling, we had a ‘solubility tag’ peptide (LGGKKKKK) synthesized and purified by GenScript. It was necessary to add an N-terminal Leu residue to the 7-residue hydrophilic tag sequence in order for the highly hydrophilic peptide to be recovered. Effect of treatment with the solubility tag peptide on activation of the NFAT reporter was assessed for both empty vector- and ∆stalkCTF-transfected cells in 3 separate signaling experiments (see figure below). Each experiment also included a negative control treatment (no peptide/culture medium only addition) and a positive control treatment (stalk peptide p17). The p17 peptide we had available was derived from the stalk sequence of human PC1 that differs from the mouse PC1 sequence at residues 15 and 17, which are two poorly conserved positions within the stalk sequence (see Reviewer 2, Response 3). In the first experiment with the solubility tag and human p17 peptides (B in figure below), we inadvertently used the empty expression vector and ∆stalkCTF expression construct from mouse PC1. After realizing our error, we then performed 2 additional signaling experiments (C and D in figure below) with the ‘correct’ human ∆stalkCTF expression construct and empty vector. In the revised manuscript, we have provided the results from each of the 3 experiments as Fig. S2 (below).

(8) (Page 9) "Using a computational model of the ΔStalk PC1 CTF developed previously". The authors might want to expand here a little to give a short review about the structure preparation.

We appreciate the reviewer’s suggestion regarding the addition of details for structure preparation for Stalkless CTF. We have added these details in section “Docking and Pep-GaMD simulations of peptide agonist binding to stalkless PC1 CTF” on Page 10 in the revised manuscript: “The cryo-EM structure of human PC1-PC2 complex (PDB: 6A70) was used to build the computational model for WT PC1 CTF. As the protein had several missing regions including the Stalk and several loops, homology modeling of the missing regions was done using I-TASSER web server. Using the WT PC1 CTF model, computational model for ΔStalk was generated by deleting the first 21 residues (3049-3069) of the WT PC1 and using the structure for stalkless CTF, we successfully docked the p9, p17 and p21 stalk peptides with HPEPDOCK. The peptides all bound to the TOP domain and the interface between the TOP domain and extracellular loop 1 (ECL1) of CTF.”

(9) How was "contact" defined when counting the number of contacts used in the 2D PMFs (Figures 2-4). Response: We appreciate the reviewer’s comment regarding the definition of the number of contacts used in the 2D PMFs. This has been clarified in the revised manuscript as: “The number of contacts is calculated between any atom pairs within 4 Å distance of the peptide and extracellular domains of PC1 protein.”(10) How was the ranking of GaMD clusters done? It looks from Figure 3A that the "intermediate" state is more favorable compared to the "bound" state, but it was claimed in the text the "bound" state was ranked 1st.

Thanks to the reviewer for this comment. It has been clarified in the revised

Supplementary Information: “Three independent Pep-GaMD simulations were combined to perform structural clustering using the hierarchical agglomerative clustering algorithm in CPPTRAJ. A 3 Å RMSD cutoff was used for each peptide system. PyReweighting was then applied to calculate the original free energy values of each peptide structural cluster with a cutoff of 500 frames. The structural clusters were finally ranked according to the reweighted free energy values.” And in the revised main text: “It is important to note that the free energy profiles calculated from GaMD simulations of PC1 CTF were not fully converged since certain variations were observed among the individual simulations. The free energy values of 2D PMF minima shown in Figure 3A could differ from those in the 1D PMF minima of peptide structural clusters, especially with the usage of distinct reaction coordinates. Nevertheless, these calculations allowed us to identify representative low-energy binding conformations of the peptides.”

(11) When mentioning residue pair distances, such as in the sentence "The distance between the TOP domain residue R3848 and PL residue E4078 was 3.8 Å (Fig. 4D)" on page 12, it should be clarified if these distances are average distance, or a statistical error can be given.

We appreciate the reviewer’s comment regarding the TOP Domain and PL distance between residues R3848-E4078. This has been clarified on page 14 in the revised manuscript as:

“The distance between the TOP domain residue R3848 and PL residue E4078 was 3.8 Å. The distance was extracted from the top-ranked structural cluster of the p21 bound to the ΔStalk CTF, corresponding to the “Closed/Active” low-energy conformational state. (Fig. 4E)”.

(12) More analysis of the GaMD can be performed. For example, the authors observed a single "bound" state for p21, but there must be some flexibility in the peptide and the protein itself. The authors might want to consider adding some plots illustrating the flexibility of the peptide residues (for example, a RMSD plot). Contact maps can also be added to visualize the results currently discussed in the text.

We thank the reviewer for their constructive suggestions. To characterize flexibility of the peptide and protein in the revised manuscript, we have added plots of the TOP-PL interaction distance between residues R3848-E4078 in PC1, the radius of gyration (Rg) of p21 and root-mean square deviation (RMSD) of p21 relative to the starting HPEPDOCK conformation of the peptide in the new Fig. S7. The peptide-protein contact map has also been added in the new Fig. S4.

(13) (Page 7) In the sentence `...sampled the "Closed/Active" low-energy state relative to the large number of Stalk-TOP contacts`, I suggest using "related to" instead of "relative to"

We thank the reviewer for the comment, and we have replaced "relative to" to “related to” in the following sentence `...sampled the "Closed/Active" low-energy state relative to the large number of Stalk-TOP contacts`

(14) (Page 7) In the sentence `Our previous study utilized expression constructs of human PC1 CTF, however, in order to prepare for ...`, "PC1 CTF, however," -> "PC1 CTF. However,"

We thank the reviewer for the comment, and we have replaced "PC1 CTF, however," to "PC1 CTF. However," in the following sentence `Our previous study utilized expression constructs of human PC1 CTF, however, in order to prepare for ...`.

**Reviewer 2:**
The autosomal dominant Polycystic Kidney Disease (ADPKD) is a major form of Polycystic Kidney Disease (PKD). To provide better treatment and avoid side effects associated with currently available options, the authors investigated an interesting GPCR, polycystin-1 (PC1), as a potential therapeutic target. *In vitro* and *in silico* studies were combined to identify peptide agonists for PC1 and to elucidate their roles in PC1 signaling. Overall, regarding the significance of the findings, this work described valuable peptide agonists for PC1 and the combined *in vitro* and *in silico* approach can be useful to study a complex system like PC1. However, the strength of the evidence is incomplete, as more experiments are needed as controls to validate the computational observations. The work appears premature.

We greatly appreciate the reviewer’s encouraging and positive comments. The reviewer’ specific comments are addressed pointwise below and changes to the text will be highlighted in yellow in the revised manuscript.

(1) The therapeutic potential of PC1 peptide agonists is unclear in the introduction. For example, while the FDA-approved drug Jynarque was mentioned, the text was misleading as it sounded like Jynarque targeted PC1. In fact, it targets another GPCR, the vasopressin receptor 2 (V2). A clear comparison of targeting PC1 over V2 pathways and their therapeutic relevance can help the readers better understand the importance of this work. Importantly, a clear background on the relationship between PC1 agonism and treatments for ADPKD is necessary.

We understand the confusion that was caused by the brevity of our introductory paragraph and will clarify the differences in therapeutic targeting between Jynarque and our PC1 stalk-derived peptides in the revised manuscript. We will also expound on the rationale for targeting PC1 agonism as a therapeutic approach for ADPKD versus Jynarque. For example: It is known that ADPKD disease severity is dependent on the functional levels of PC1. Jynarque is a small molecule antagonist of the arginine vasopressin receptor 2, V2R, whose signaling, and production of cAMP has been shown to be increased in ADPKD. As this drug targets one of the downstream aberrant pathways, it is only capable of slowing disease progression and has numerous undesirable side effects. We reasoned that a therapeutic agent capable of stimulating and thus augmenting PC1 signaling function would be a safer, cyst initiation-proximal treatment capable of preventing cyst formation with few side effects.

(2) PC1 is a complex membrane protein, and most figures focus on the peptide-binding site. For general readers (or readers that did not read the previous PNAS publication), it is hard to imagine the overall structure and understand where the key interactions (e.g., R3848-E4078) are in the protein and how peptide binding affects locally and globally. I suggest enhancing the illustrations.

We thank the reviewer for the constructive comment on adding more illustrations for the PC1 protein to understand the overall structure and the location of the key interaction R3848E4078. We have included these suggestions and modified the main figures in the revised manuscript.

(3) The authors used the mouse construct for the cellular assays and the peptide designs in preparation for future in vivo assays. This is helpful in understanding biology, but the relevance of drug discovery is weakened. Related to Point 1, the therapeutic potential of PC1 peptide agonist is largely missing.

The therapeutic potential of a PC1 peptide agonist is addressed in response #1 above. As mentioned in the manuscript and recognized by the reviewer, the cellular signaling assays were performed with the mouse PC1 CTF expression construct and with peptides based on the mouse PC1 stalk sequence for future, pre-clinical studies, while the peptide binding studies were performed with the human PC1 stalk sequence. We feel the relevance for drug discovery is not significantly weakened for a number of reasons: (1) as shown in Fig. 1A, the stalk sequence is highly conserved between mouse and human PC1, specifically there are only 2 residue differences present within peptides p17 and p21. One of the differences is a ‘semi-conservative’ Gln-Arg substitution at peptide residue 15, while the second difference is a conservative Ile-Val substitution at peptide residue 17; (2) we have found that an Arg to Cys mutation within the mouse PC1 CTF stalk has the same effect on signaling as the corresponding human Gln to Cys ADPKD-associated mutation which was analyzed in (Pawnikar et al., 2022); and (3) both peptide residues 15 and 17 represent highly variable positions within the PC1 stalk as shown in the sequence logo (below) of the stalk sequence from 16 vertebrate species; and (4) while addressing the potential effect of the hydrophilic solubility tag on stalk peptide-mediated rescue of CTF∆stalk signaling (see Reviewer 1 comments, point #7), we utilized the ‘human’ version of p17 as a positive control and tested its activation with both mouse and human CTF∆stalk expression constructs and found that human p17 peptide was also capable of stimulating the mouse CTF∆stalk protein (Fig. S2).

**Author response image 1. sa4fig1:** 

(4) More control experiments are needed. For example, a 7-residue hydrophilic sequence (GGKKKKK) is attached to the peptide design to increase solubility. This 7-residue peptide should be tested for PC1 activation as a control. Second, there is no justification for why the peptide design must begin with residue T3041. Can other segments of the stalk also be agonists?

As mentioned above for Reviewer 1, the hydrophilic peptide has been synthesized and tested for activation of signaling by the stalkless CTF in the revised manuscript as Fig. S2. The design of peptides that begin with residue T3041 of mouse PC1 CTF is modeled on numerous similar studies for the family of adhesion GPCRs. Optimization of the binding and activity of the PC1 peptide agonist will be investigated in future studies and could include such parameters as whether the peptide must include the first residue and whether subsegments of the stalk are also agonists, however, we feel these questions are beyond the scope of this initial report.

(5) There are some major concerns about the simulations: The GaMD simulations showed different binding sites of p-21, p-17, and p-9, and the results report the simulated conformations as "active conformational states". However, these are only computational findings without structural biology or mutagenesis data to validate. Further, neither docking nor the simulation data can explain the peptide SAR. Finally, it will be interesting if the authors can use docking or GaMD and explain why some peptide designs (like P11-P15) are less active (as control simulations).

The reviewer brings up an important observation regarding differences in binding sites between peptides p9, p17 and p21. We will include discussion of this observation and our interpretations to the revised manuscript. While the present study is focused on identification of initial peptides that are able to activate the PC1 CTF, we shall include further mutation experiments and simulations, peptide SAR and optimization of the lead peptides in future studies. This has been clarified in the revised manuscript.

(6) Additional experiments for the controls and for validating the simulations. Additional simulations to explain the SAR.

We appreciate the reviewer’s comment for additional experiments for the controls and additional simulations to explain the SAR. For future studies, we shall include further mutation experiments and simulations, peptide SAR and optimization of the lead peptides.

(7) What is the selectivity of the peptides between PC1 and PC2?

We have not tested the selectivity of the peptides for PC1 versus PC2 primarily because transfection of PC2 does not activate the NFAT reporter. However, it is possible that co-transfection of PC2 with the PC1 CTF could alter stalk peptide binding. This will be important to consider in future studies.

**Reviewer 3:**
The authors demonstrate the activation of polycystin-1 (PC1), a G-protein coupled receptor, using small peptides derived from its original agonist, the stalk TA protein. In the experimental part of the study, the authors performed cellular assays to check the peptide-induced reactivation of a mutant form of PC1 which does not contain the stalk agonist. The experimental data is supported by computational studies using state-of-the-art Gaussian accelerated Molecular Dynamics (GaMD) and bioinformatics analysis based on sequence covariance. The computer simulations revealed the mechanistic details of the binding of the said peptides with the mutant PC1 protein and discovered different bound, unbound, and intermediate conformations depending on the peptide size and sequence. The use of reliable and well-established molecular simulation algorithms and the physiological relevance of this protein autosomal dominant Polycystic Kidney Disease (ADPKD) make this work particularly valuable.

We greatly appreciate the reviewer’s encouraging and positive comments. The reviewer’ specific comments are addressed pointwise below and changes to the text will be highlighted in yellow in the revised manuscript.

(1) No control has been used for the computational (GaMD) study as the authors only report the free energy surface for 3 highly agonistic peptides but for none of the other peptides that did not induce an agonistic effect. Therefore, in the current version, the reliability of the computational results is not foolproof.

We appreciate the reviewer’s concern about the lack of control with the other peptides that did not induce an agonistic effect. To address the reviewer’s concern, we have included more details on the study of the stalkless CTF and the solubility tag peptide (Fig. S2) as controls in the revised manuscript.

(2) All discussions about the residue level interactions focused only on geometric aspects (distance, angle, etc) but not the thermodynamic aspect (e.g. residue-wise interaction energy). Considering they perform a biased simulation; the lack of interaction energy analysis only provides a qualitative picture of the mechanism.

As mentioned by the reviewer, we have added MM/PBSA analysis results in the revised manuscript and SI.

Molecular Mechanics/Poisson-Boltzmann Surface Area (MM/PBSA) analysis was performed to calculate the binding free energies of peptides p9, p17 and p21 to PC1 CTF. The analysis was performed using the trajectory in which the peptide was bound to the receptor. In MM/PBSA, the binding free energy of the ligand (L) to the receptor (R) to form the complex (RL) is calculated as:

ΔGbind=GRL−GR−GL

where GRL is the Gibbs free energy of the complex RL, GR is the Gibbs free energy of the molecule R in its unbound state and GL is the Gibbs free energy of the molecule L in its unbound state, respectively.

𝛥𝐺𝑏𝑖𝑛𝑑 can be divided into contributions of different interactions as:

ΔGbind=ΔH−TΔS=ΔEMM+ΔGsol −TΔS

in which

ΔGsol =ΔGPB/GB+ΔGSA

ΔGSA=γ⋅SASA+b

where ΔEMM , ΔGsol , 𝞓H and −TΔS are the changes in the gas-phase molecular mechanics (MM) energy, solvation free energy, enthalpy and conformational entropy upon ligand binding, respectively. ΔEMM includes the changes in the internal energies ΔEint (bond, angle and dihedral energies), electrostatic energies ΔEelec , and the van der Waals energies ΔEvdW. ΔGsol is the sum of the electrostatic solvation energy ΔGPB/GB (polar contribution) and the nonpolar contribution ΔGSA between the solute and the continuum solvent. The polar contribution is calculated using either the Poisson Boltzmann (PB) or Generalized Born (GB) model, while the nonpolar energy is usually estimated using the solvent-accessible surface area (SASA) where 𝞬 is surface tension coefficient and b is the constant offset. The change in conformational entropy −TΔS is usually calculated by normal-mode analysis on a set of conformational snapshots taken from MD simulations. However, due to the large computational cost, changes in the conformational entropy are usually neglected as we were concerned more on relative binding free energies of the similar peptide ligands.

MM/PBSA analysis was performed using the gmx_MMPBSA software with the following command line:

gmx_MMPBSA -O -i mmpbsa.in -cs com.tpr -ci index.ndx -cg 1 13 -ct com_traj.xtc -cp topol.top -o FINAL_RESULTS_MMPBSA.dat -eo FINAL_RESULTS_MMPBSA.csv Input file for running MM/PBSA analysis:

&generalsys_name="Prot-Pep-CHARMM",startframe=1, endframe=200, # In gmx_MMPBSA v1.5.0 we have added a new PB radii set named charmm_radii.# This radii set should be used only with systems prepared with CHARMM force fields.# Uncomment the line below to use charmm_radii set# PBRadii=7,

/

&pb

# radiopt=0 is recommended which means using radii from the prmtop file for both the PB calculation and for the NP

# calculation

istrng=0.15, fillratio=4.0, radiopt=0

The relative rank of the overall peptide binding free energies (Table S1) was consistent with the experimental signaling data, i.e., p21>p9>p17, for which p21 showed the largest binding free energy value of binding (-40.29±6.94 kcal/mol).

(3) It is not mentioned clearly whether the reader should interpret the free energy landscapes quantitatively or qualitatively. Considering no error analysis or convergence plots are reported for the GaMD free energy surfaces, it may be assumed the results are qualitative. The readers should consider this caveat and not try to quantitatively reproduce these free energy landscapes with other comparable techniques.

We appreciate the reviewer’s comment whether the free energy landscapes should be interpreted quantitatively or qualitatively. The presented free energy landscapes could be considered semi-quantitative since the simulations are not fully converged. This will be clarified in the revised manuscript as: “It is important to note that the free energy profiles calculated from GaMD simulations of PC1 CTF were not fully converged since certain variations were observed among the individual simulations. Nevertheless, these calculations allowed us to identify representative low-energy binding conformations of the peptides.”

(4) Energy decomposition analysis similar to the following paper (https://pubs.acs.org/doi/10.1021/bi201856m) should be provided to understand the residue level enthalpic contribution in the peptide-protein interaction.

As mentioned by the reviewer, we have performed residue-wise interaction energy analysis and included the analysis results in the revised manuscript and SI.

Residue-wise interaction energy analysis was performed on peptides p9, p17 and p21 using the trajectory in which the peptide was bound to the PC1 CTF using the gmx_MMPBSA software with the following command line:

gmx_MMPBSA -O -i mmpbsa.in -cs com.tpr -ct com_traj.xtc -ci index.ndx -cg 3 4 -cp topol.top -o FINAL_RESULTS_MMPBSA.dat -eo FINAL_RESULTS_MMPBSA.csv -do FINAL_DECOMP_MMPBSA.dat -deo FINAL_DECOMP_MMPBSA.csv

Input file for running residue-wise energy decomposition analysis:

&generalsys_name="Decomposition", startframe=1, endframe=200,# forcefields="leaprc.protein.ff14SB"

/

&gb

igb=5, saltcon=0.150,

/

# make sure to include at least one residue from both the receptor #and peptide in the print_res mask of the &decomp section.

# this requirement is automatically fulfilled when using the within keyword.

# http://archive.ambermd.org/201308/0075.html

&decomp

idecomp=2, dec_verbose=3, print_res="A/854-862 A/1-853”,

/

Residue-wise energy decomposition analysis allowed us to identify key residues that contributed the most to the peptide binding energies. These included residues T1 and V9 in p9 (Table S2), residues T1, R15 and V17 in p17 (Table S3), and residues P10, P11, P19 and P21 in p21 and residue W3726 in the PC1 CTF (Table S4). The energetic contributions of these residues apparently correlated to the sequence coevolution predicted from the Potts model.

(5) To showcase the reliability of the computational approach, the authors should perform the MD simulation studies with one peptide that did not show any significant agonistic effect in the experiment. This will work as a control for the computational protocol and will demonstrate the utility of the pep-GaMD simulation in this work.

We appreciate the reviewer’s concern about the lack of control with the other peptides that did not induce an agonistic effect. It is difficult for us to add more MD simulations on the other peptides, due to student leave after PhD graduation. But to address the reviewer’s concern, we have included more details on the study of the stalkless CTF as a control in the revised manuscript.

(6) To assess the accuracy of the computational results the authors should mention (either in the main text or SI) whether the reported free energy surfaces were the average of the five simulations or computed from one simulation. In the latter case, free energy surfaces computed from the other four simulations should be provided in the SI. In addition, how many binding unbinding events have been observed in each simulation should be mentioned.

We appreciate the reviewer’s comment regarding convergence of the simulation free energy surfaces. In response to Reviewer 1, we have calculated free energy profiles of individual simulations for each system, including the p9, p17, and p21 (Figs. S5, S6 and S8).

“We have calculated free energy profiles of individual simulations for each system, including the p9, p17, and p21 (Figs. S5, S6 and S8). For the p9 peptide, the “Bound” low-energy state was consistently identified in the 2D free energy profile of each individual simulation (Fig. S5). For the p17 peptide, Pep-GaMD simulations were able to refine the peptide conformation from the "Unbound” to the "Intermediate” and “Bound” states in Sim1 and Sim5, while the peptide reached only the "Intermediate” state in the other three simulations (Fig. S6). For the p21 peptide, PepGaMD was able to refine the peptide docking conformation to the "Bound” state in all the five individual simulations (Fig. S8).”

“It is important to note that the free energy profiles calculated from GaMD simulations of PC1 CTF were not fully converged since certain variations were observed among the individual simulations. Nevertheless, these calculations allowed us to identify representative low-energy binding conformations of the peptides.”